# Cell-density independent increased lymphocyte production and loss rates post-autologous HSCT

Mariona Baliu-Piqué[1†], Vera van Hoeven[2†], Julia Drylewicz[1†], Lotte E van der Wagen[3], Anke Janssen[1], Sigrid A Otto[1], Menno C van Zelm[4], Rob J de Boer[5], Jürgen Kuball[1,3], Jose AM Borghans[1*], Kiki Tesselaar[1*]

[1]Center for Translational Immunology, University Medical Center Utrecht, Utrecht, Netherlands; [2]Department of Experimental Immunology, Amsterdam UMC, University of Amsterdam, Amsterdam, Netherlands; [3]Department of Hematology, University Medical Center Utrecht, Utrecht, Netherlands; [4]Department of Immunology and Pathology, Monash University and Alfred Hospital, Melbourne, Australia; [5]Theoretical Biology, Utrecht University, Utrecht, Netherlands

**Abstract** Lymphocyte numbers need to be quite tightly regulated. It is generally assumed that lymphocyte production and lifespan increase homeostatically when lymphocyte numbers are low and, vice versa, return to normal once cell numbers have normalized. This widely accepted concept is largely based on experiments in mice, but is hardly investigated in vivo in humans. Here we quantified lymphocyte production and loss rates in vivo in patients 0.5–1 year after their autologous hematopoietic stem cell transplantation (autoHSCT). We indeed found that the production rates of most T- and B-cell subsets in autoHSCT-patients were two to eight times higher than in healthy controls, but went hand in hand with a threefold to ninefold increase in cell loss rates. Both rates also did not normalize when cell numbers did. This shows that increased lymphocyte production and loss rates occur even long after autoHSCT and can persist in the face of apparently normal cell numbers.

**\*For correspondence:**
J.Borghans@umcutrecht.nl (JAMB);
K.Tesselaar@umcutrecht.nl (KT)

[†]These authors contributed equally to this work

## Introduction

Under healthy conditions, the peripheral T- and B-cell populations are maintained at relatively constant numbers throughout life (*Lin et al., 2016*; *Wertheimer et al., 2014*). Homeostatic mechanisms are thought to regulate lymphocyte production and survival rates in a density-dependent manner. Indeed, studies in rodents have shown that lymphocyte division and lifespan increase in response to severe lymphopenic conditions (*Freitas and Rocha, 2000*). Robust peripheral proliferation of T-cells occurs both upon adoptive cell transfer into severely lymphocyte-depleted mice and in partially immune-depleted hosts in the absence of adoptive cell transfer, a phenomenon termed lymphopenia-induced proliferation (LIP) (*Freitas and Rocha, 2000*; *Miller and Stutman, 1984*; *Bell et al., 1987*; *Neujahr et al., 2006*). Similarly, rapid proliferation and extended survival of B-cells occur after adoptive cell transfer into B-cell deficient hosts and correlate with peripheral B-cell numbers (*Gaudin et al., 2004*).

By analogy, it is generally assumed that lymphopenic conditions induce alterations in lymphocyte dynamics in humans. However, in humans full recovery of the T-cell compartment following an autologous hematopoietic stem cell transplantation (autoHSCT) is notoriously slow, often taking several years (*Heining et al., 2007*; *Ringhoffer et al., 2013*; *Bosch et al., 2012*; *Williams et al., 2007*). On the basis of elevated frequencies of Ki-67[+] cells, severe lymphopenia arising after HSCT and lymphocyte-depleting treatments has been associated with increased proliferation of naive and memory

T-cells (*Jones et al., 2013*; *Bouvy et al., 2013*; *Hazenberg et al., 2002*; *Alho et al., 2016*). However, elevated frequencies of Ki-67[+] cells were shown to decline within 3–6 months after cell depletion, despite the fact that patients were still deeply lymphopenic (*Bouvy et al., 2013*; *Hazenberg et al., 2002*; *Alho et al., 2016*). Furthermore, increased T-cell proliferation rates after allogeneic HSCT have been shown to correlate with the occurrence of graft-versus-host disease (GVHD) and infectious-disease-related complications (*Hazenberg et al., 2002*). Together, these observations question whether homeostatic mechanisms are induced to compensate for low lymphocyte numbers in humans undergoing HSCT. It remains unclear to what extent increased T-cell proliferation post-HSCT reflects a T-cell density-dependent response to lymphopenia, or an immune response triggered by therapy-related tissue damage, infectious complications or immune activation.

To elucidate whether lymphocyte production and death rates in humans are regulated in a density-dependent manner, we used in vivo deuterium labeling to quantify the production and loss rates of different T- and B-cell subsets in patients who received an autologous HSCT (autoHSCT), and had no signs of clinically manifested infections or GVHD. Twelve months after autoHSCT, absolute numbers of CD4[+] T-cells and memory and natural effector B-cells in these patients were still lower than in healthy individuals, while CD8[+] T-cell and naive B-cell numbers had already recovered to healthy control (HC) values. Deuterium labeling revealed that the production rates of most lymphocyte subsets, even those that had already reconstituted, were significantly higher in patients post-autoHSCT than in healthy individuals. These increased rates of T- and B-cell production could only be reconciled with the observed stable cell numbers over the study time if lymphocyte loss rates were also significantly increased. Our data therefore show that increased lymphocyte production and loss rates occur long after autoHSCT, and can even persist when a lymphocyte subset has already normalized.

## Results

### Heterogeneous T-cell reconstitution kinetics post-autoHSCT

To investigate whether lymphocyte production and loss depend on cell numbers during lymphopenia in humans, we quantified the production and loss rates of T- and B-cells in six patients who received an autoHSCT for the treatment of hematological malignancies. Patients were included in the study between 196 days and 420 days post-autoHSCT, received deuterated water ($^2H_2O$) for 6 weeks, and were followed for approximately 1 year after start of the labeling period (*Figure 1*).

*Patient B* withdrew from the study 10 weeks after the start of $^2H_2O$ labeling due to infectious complications unrelated to participation in the study. All other patients had no complications that needed treatment during the study follow-up, which was supported by C-reactive protein (CRP) levels in the normal range (*Figure 2*).

The sub-optimal T-cell recovery observed in the peripheral blood of patients post-autoHSCT (*Figure 3A*) was largely due to the slow reconstitution of CD4[+] T-cells (*Figure 3B*). At the start of $^2H_2O$ labeling, CD8[+] T-cell numbers had reached normal levels in most patients, whereas CD4[+] T-cell numbers remained below normal levels even 1.5 years post-autoHSCT. This resulted in an inverse CD4:CD8 ratio in all patients except for *patient C* (*Figure 3C*), who experienced extremely slow CD8[+] T-cell reconstitution (*Figure 3D*). Naive (CD45RO[-]CD27[+]) CD4[+] T-cell numbers remained below normal levels throughout the 2-year follow-up period, whereas memory (CD45RO[+]) CD4[+] T-cells reached the lower range of normal levels around 400 days post-autoHSCT (*Figure 3B*). Naive and memory CD8[+] T-cell numbers were at normal or supra-normal levels at the start of the study in all patients except for *patient C* (*Figure 3D*). In line with cell numbers, for most patients the fractions of naive cells, central memory (CM, CD45RO[+]CD27[+]), effector memory (EM, CD45RO[+]CD27[-]), and effector (CD45RO[-]CD27[-]) T-cells differed from those in HCs and varied slightly over time (*Figure 3E* and *Figure 3—figure supplement 1*).

Because it is generally assumed that during lymphopenia the availability of growth and survival factors increases, which has in particular been shown for IL-7 plasma levels (*van Gent et al., 2011*; *Sauce et al., 2012*; *Fry et al., 2001*; *Napolitano et al., 2001*; *Bolotin et al., 1999*), we also determined plasma levels of IL-7 and IL-15 between 12 and 24 months post-autoHSCT. Despite the CD4[+] T-cell lymphopenia observed in these patients, their plasma concentrations of IL-7 and IL-15 and several other cytokines were in the range of those of HCs (*Figure 2*).

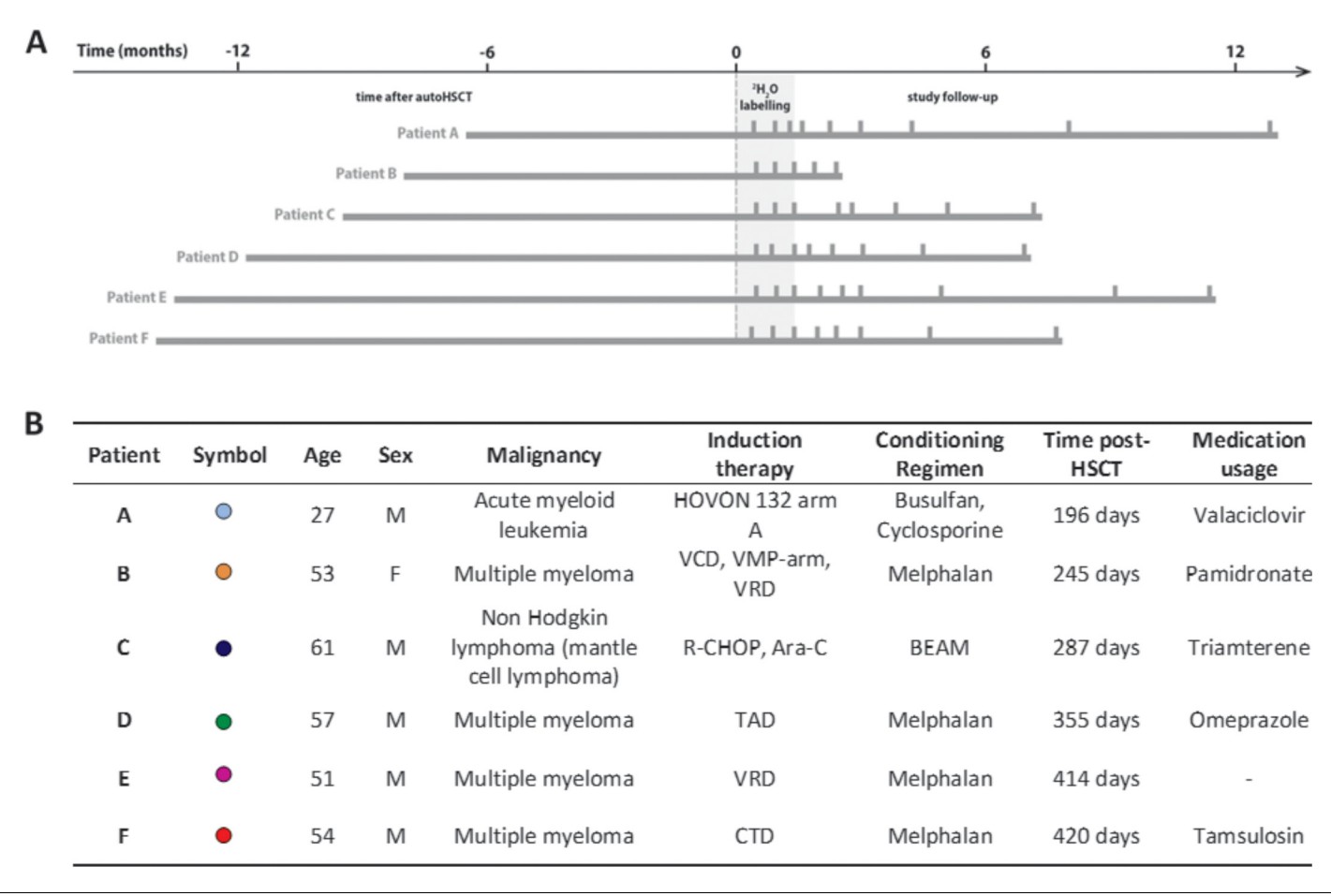

**Figure 1.** Study protocol timeline and patient characteristics. (**A**) Summary figure depicting the study time line of every patient. Patients are centered by start of $^2H_2O$ labeling. The left bar indicates the time between the autologous hematopoietic stem cell transplantation (autoHSCT) and the start of the labeling period, the gray area indicates the 6 weeks $^2H_2O$ labeling period, the right bar provides the follow-up period, and the vertical bars indicate the blood sampling time points. (**B**) Patient characteristics. Age=age at start $^2H_2O$ labeling; M=male; F=female; Time post-HSCT=reconstitution period at start $^2H_2O$ labeling; Medication usage=medication during the study; HOVON 132 arm A (Idarubicin, Ara-C (Cytarabine), Daunorubicin); VCD (Bortezomib, Cyclophosphamide, Dexamethason); VMP (Bortezomib, Melphalan, Prednisone); VRD (Bortezomib, lenalidomide, dexamethasone); R-CHOP (Rituximab, Cyclophosphamide, Adriamycin, Vincristin, Prednisone); Ara-C (Cytarabine), TAD (Thalidomide, Adriamycin, Dexamethasone); CTD (Carfilzomib, Thalidomide, Dexamethasone); BEAM (Carmustine, Etoposide, Ara-C [Cytarabine], Melphalan). *Figure 1—figure supplement 1* shows the absolute leukocytes, neutrophils, lymphocytes, and monocytes numbers over time after autoHSCT.

The online version of this article includes the following figure supplement(s) for figure 1:

**Figure supplement 1.** Absolute numbers (cells × 10^9 per liter of blood) of leukocytes, neutrophils, lymphocytes, and monocytes in peripheral blood of autologous hematopoietic stem cell transplantation (autoHSCT) patients (patients A–F) over time after autoHSCT obtained by automated blood leukocyte differential count (leukodiff).

## Increased CD4+ and CD8+ T-cell production rates post-autoHSCT

To investigate whether low CD4+ T-cell numbers were associated with increased T-cell production rates, we compared the level of deuterium enrichment in the DNA of the different T-cell subsets between patients and controls. Deuterium enrichment analysis showed a relatively high level of label incorporation in patients, despite the fact that the labeling period was 3 weeks shorter for patients than for controls (*Figure 4A*). Using mathematical modeling (see Materials and methods section) we estimated the production rates of the different T-cell subsets (i.e. the number of new cells produced per day, coming from a source or peripheral cell division, divided by the number of resident cells in the population). We found that the production rates of naive and memory CD4+ T-cells were, respectively, six times and three times higher in patients than in controls. For naive and memory

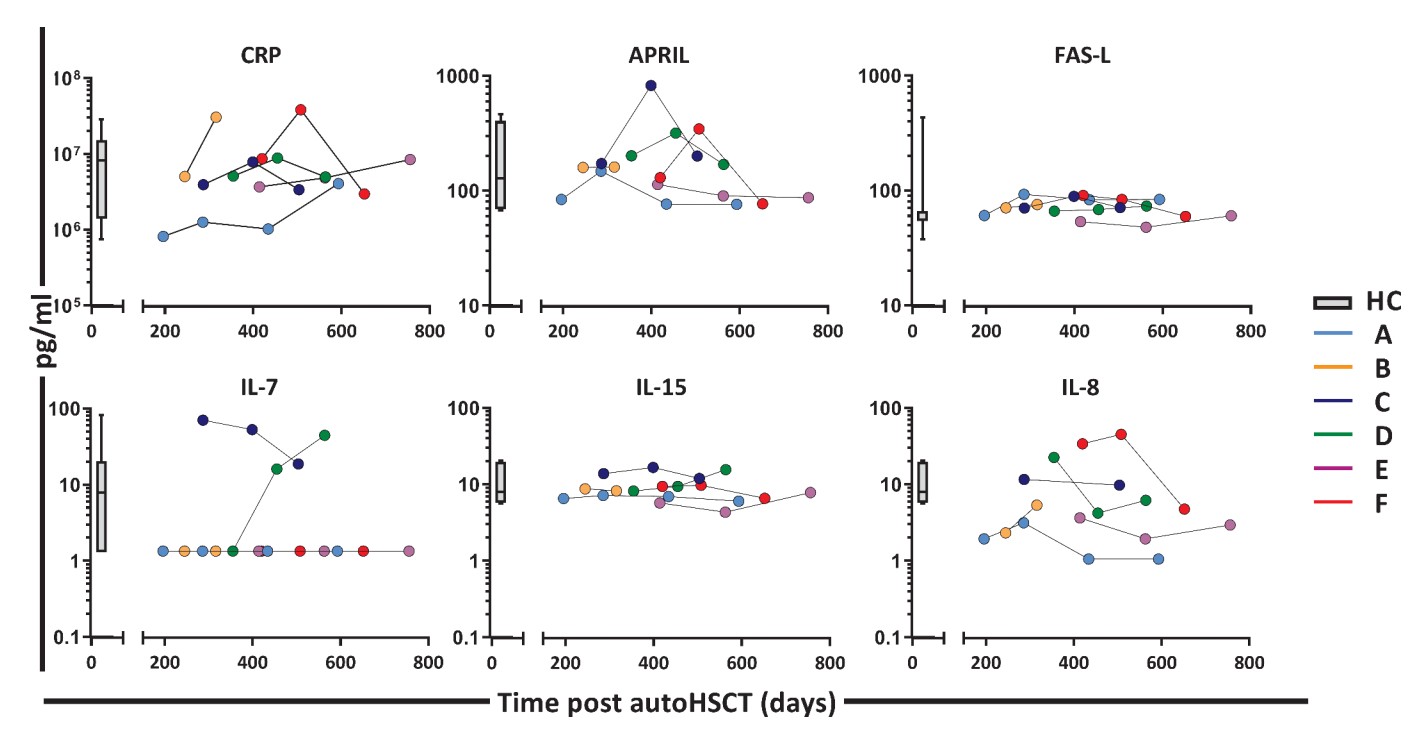

**Figure 2.** Plasma levels of CRP, APRIL, FAS-L, IL-7, IL-15, and IL-8 post-autologous hematopoietic stem cell transplantation (autoHSCT). Plasma concentration (picogram per milliliter) of CRP (C-reactive protein), APRIL (A proliferation-inducing ligand), FAS-L (FAS ligand), IL-7 (Interleukin 7), IL-15 (Interleukin 15), and IL-8 (Interleukin 8) in patients A–F at different time points after autoHSCT. Box plots represent the distribution of values for healthy controls (n = 29, box = 25th to 75th percentile, black line=median, whiskers=min and max values). IL-7 levels which were below the level of detection (1.3 pg/ml) were set at 1. *Figure 2—source data 1* shows the individual values of the different plasma markers.
The online version of this article includes the following source data for figure 2:

**Source data 1.** Luminex data for different plasma markers, time points and patients.

CD8[+] T-cells, the estimated production rates were approximately eight and four times higher in patients compared to controls (*Figure 4B* and *Figure 4—source data 2*), despite the fact that absolute CD8[+] T-cell numbers had already recovered to healthy levels 12 months post-transplantation.

## Increased proliferation of naive but not memory CD4[+] and CD8[+] T-cells

T-cell production rates as measured by deuterium labeling may reflect proliferation (i.e. either occasional self-renewal or a continuous burst of cell division) of the subset of interest or an influx of cells from a source (e.g. by thymic output) or from another subset (e.g. through lymphocyte differentiation). To distinguish between these options, we first measured Ki-67 expression, a snapshot marker of recent proliferative activity which, in contrast to deuterium labeling, allows to distinguish between cell division and influx. The fraction of Ki-67[+] cells within the naive CD4[+] and CD8[+] T-cell pools was significantly higher in patients compared to controls (*Figure 5A*). For the memory T-cell subsets, in contrast, the fraction of Ki-67[+] cells of patients did not differ significantly from those of controls (*Figure 5A*). This suggests that the increased production rates of memory CD4[+] and CD8[+] T-cells may occur due to an increased influx from naive T-cells into the memory compartment, rather than increased T-cell division within the memory T-cell pools.

Besides increased cell division in the naive T-cell pool, increased naive T-cell production rates post-autoHSCT could in theory also be due to increased thymic output. T-cell receptor excision circles (TRECs) are commonly measured to estimate thymopoiesis. Because the average TREC content per T-cell declines with age (*Douek et al., 1998*; *Hazenberg et al., 2000*; *Hazenberg et al., 2001*), we measured TREC contents of naive T-cells from patients, cord blood (CB), and young (on average 23 years of age) and aged (on average 68 years of age) healthy individuals (*Westera et al., 2015*). The average TREC content of naive CD4[+] T-cells in patients was approximately 10-fold higher

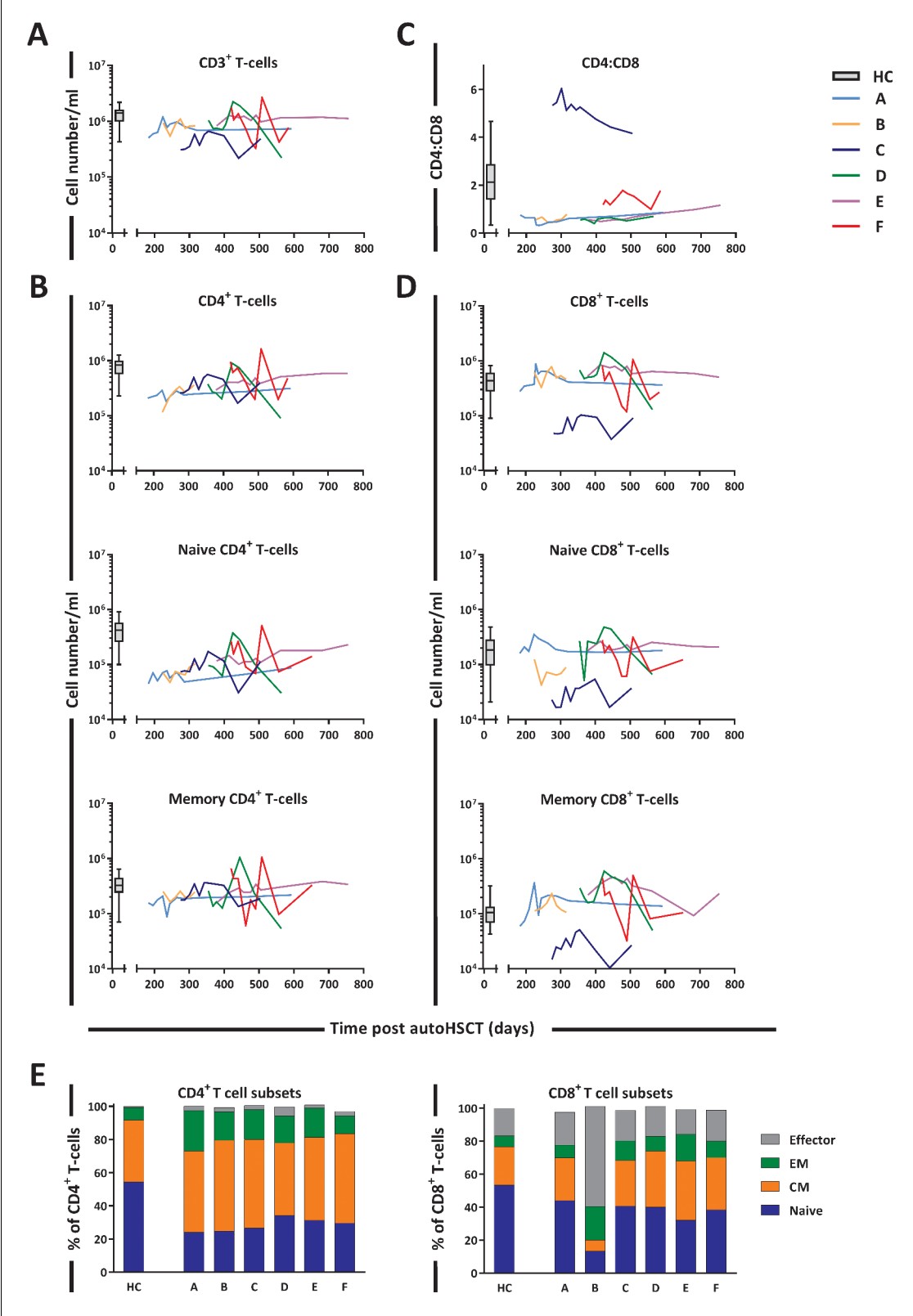

**Figure 3.** T-cell reconstitution following autologous hematopoietic stem cell transplantation (autoHSCT). Absolute numbers (cells per milliliter) of (**A**) total CD3[+] T-cells, (**B**) total, naive (CD27[+]CD45RO[-]) and memory (CD45RO[+]) CD3[+]CD4[+] T-cells, (**D**) total, naive (CD27[+]CD45RO[-]) and memory (CD45RO[+]) CD3[+]CD8[+] T-cells, and (**C**) the CD4:CD8 ratio over time for the duration of the study are depicted. Box plots represent the distribution of values for healthy controls (HCs) (N = 17 for CD3[+], CD4[+], CD8[+], and CD4:CD8 ratio, other N = 27, box = 25th to 75th percentile, black line=median,

*Figure 3 continued on next page*

*Figure 3 continued*

whiskers=min and max values). Absolute numbers shown in the graph are not normalized. (E) Bar graphs show the median percentage of naive (CD27[+]CD45RO[-]), central memory (CM, CD27[-]CD45RO[+]), effector memory (EM, CD27[+]CD45RO[+]), and effector (CD27[-]CD45RO[-]) CD4[+] and CD8[+] T-cells of autoHSCT patients (A–F) and HCs (n = 6) in the indicated colors. For the T-cell subset distribution per patient over time, see *Figure 3—figure supplement 1* and *Figure 3—source data 1*. For the gating strategy, see *Figure 3—figure supplement 2*.

The online version of this article includes the following source data and figure supplement(s) for figure 3:

**Source data 1.** T-cell numbers and percentages in blood of patients and healthy controls.
**Figure supplement 1.** T-cell subset distribution per autologous hematopoietic stem cell transplantation (autoHSCT) patient over time.
**Figure supplement 2.** Gating strategy for TruCount analysis and cell sorting.
**Figure supplement 3.** Total daily production of T- and B-cell subsets, calculated as average production rate $\left(\frac{\alpha}{N} + p_N\right) \times$ (absolute number of cells per liter of blood) × (five liter blood) × 50, assuming that 2% of lymphocytes reside in the blood (*Ponchel et al., 2011*).

than in aged controls, and not significantly different from that of young individuals and CB (*Figure 5B*), even though all but one of the patients were more than 50 years of age. For naive CD8[+] T-cells, the average TREC content in patients was in the range of young and aged controls (*Figure 5B*). We also measured CD31 expression on naive CD4[+] T-cells, as CD31[+]CD4[+] T-cells are known to be enriched in recent thymic emigrants (RTEs) (*Kohler and Thiel, 2009*; *van den Broek et al., 2018*). The fraction of CD31[+] cells within the naive CD4[+] T-cell population was slightly higher in patients than in aged controls and slightly lower than in young controls and CB (*Figure 5C* and *Figure 5—figure supplement 2*). For naive T-cells, the combined Ki-67, TREC, and CD31 data suggest that the increased T-cell production rate post-HSCT is at least partially due to increased T-cell division. Since the increased average TREC contents and percentages of CD31[+] cells may be a direct consequence of normal thymic output entering a smaller T-cell pool (*Hazenberg et al., 2003*), the contribution of thymic output to the increased T-cell production rate in these patients remains unclear.

## Heterogeneous B-cell reconstitution kinetics post-autoHSCT
Next, we studied the changes in B-cell dynamics following autoHSCT. Although total CD19[+] B-cell numbers and naive (IgM[+]CD27[-]) B-cell numbers had already reached normal or even supra-normal levels by day 200 post-autoHSCT, Ig class-switched (IgM[-]CD27[+]) and IgM[+] (IgM[+]CD27[+]) memory B-cell numbers in most patients were still below, or in the lower range of, those of HCs throughout the study period (*Figure 6A and B* and *Figure 6—figure supplement 1*).

## Increased production rates of B-cells post-autoHSCT
We analyzed the deuterium enrichment of the different B-cell subsets to study whether B-cell production rates were increased for subsets which were still low in cell numbers (*Figure 7A*). The production rates of Ig switched-memory B-cells and IgM[+] memory B-cells were 3.5 times and 5 times higher than in controls, respectively (*Figure 7B* and *Figure 7—source data 1*). Also the production rate of naive B-cells, a population that had already reconstituted to supra-normal levels, remained significantly higher than in HCs (*Figure 7B*, *Figure 7—source data 1*).

Because B-cell production may depend on peripheral B-cell division and on de novo bone marrow output, we measured Ki-67 expression and kappa-deleting recombination excision circles (KRECs), in an attempt to estimate bone marrow output. The percentages of dividing, that is, Ki-67[+], cells within IgM[+] and Ig class-switched memory B-cells were significantly higher in patients than in healthy individuals (*Figure 7C*). In contrast, the fraction of Ki-67[+] cells within the naive B-cell subset was similar between patients and controls (*Figure 7C*). Although naive B-cell peripheral division rates were not increased post-autoHSCT, their production rates were two times higher than in controls. The division history (measured as number of cell divisions) of the naive B-cell subsets in patients tended to be lower than in controls (although not significantly), suggesting that the output of naive B-cells from the bone marrow rather than their peripheral proliferation rate was increased after autoHSCT (*Figure 7D*).

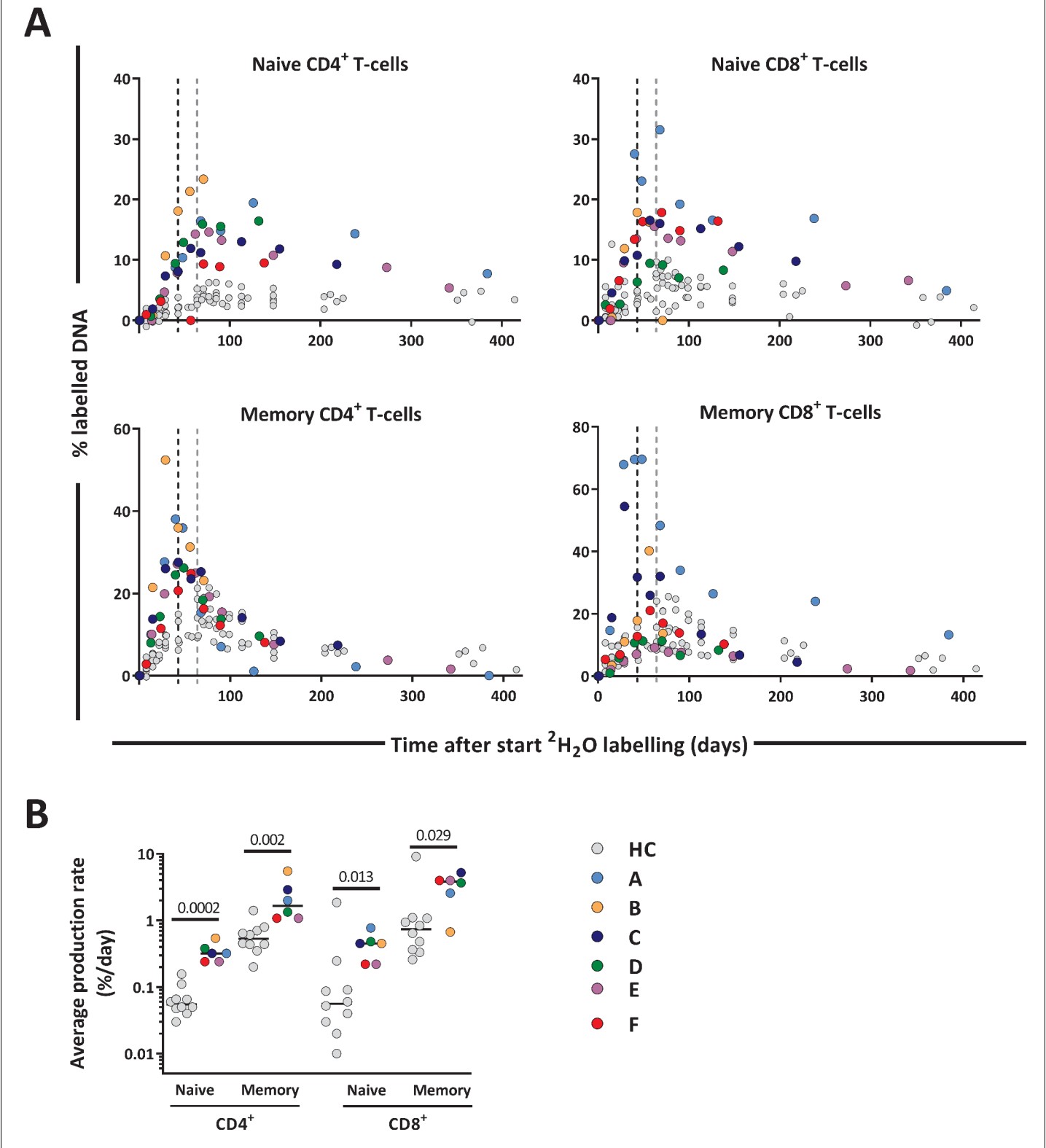

**Figure 4.** T-cell dynamics after autologous hematopoietic stem cell transplantation (autoHSCT). (**A**) Deuterium enrichment in the DNA of naive and memory CD4[+] and CD8[+] T-cells in autoHSCT patients (A–F, color symbols) and healthy controls (HCs, gray symbols) (***Westera et al., 2015***). Dotted lines correspond to the end of the labeling period (black for autoHSCT patients and gray for HCs). Label enrichment was scaled between 0% and 100% by normalizing for the maximum enrichment in granulocytes (see ***Figure 4—figure supplement 1*** and ***Figure 4—source data 1***). (**B**) Estimates of the

*Figure 4 continued on next page*

*Figure 4 continued*

per cell production rate of naive and memory CD4[+] and CD8[+] T-cells in autoHSCT patients and HCs (*Westera et al., 2015*) (for individual fits and parameters estimates, see *Figure 4—figure supplement 2* and *Figure 4—source data 2*). Different symbols indicate different individuals, autoHSCT patients (A–F) in color, and HCs in gray. Horizontal lines represent median values. p-values between groups are shown (Mann–Whitney test). For information on modeling in R, see *Figure 4—source data 1*.

The online version of this article includes the following source data, source code and figure supplement(s) for figure 4:

**Source code 1.** Modeling in R and raw data.

**Source data 1.** Estimates of urine and granulocyte parameters and their corresponding 95% confidence limits for deuterium enrichment.

**Source data 2.** Estimates of average daily production rates for T-cell subsets of autoHSCT patients.

**Figure supplement 1.** Best fits of $^2$H enrichment in (A) body water (urine) and (B) granulocytes from the six autologous hematopoietic stem cell transplantation (autoHSCT) patients (A–F).

**Figure supplement 2.** Best fits of $^2$H enrichment in T-cell subsets in autologous hematopoietic stem cell transplantation (autoHSCT) patients.

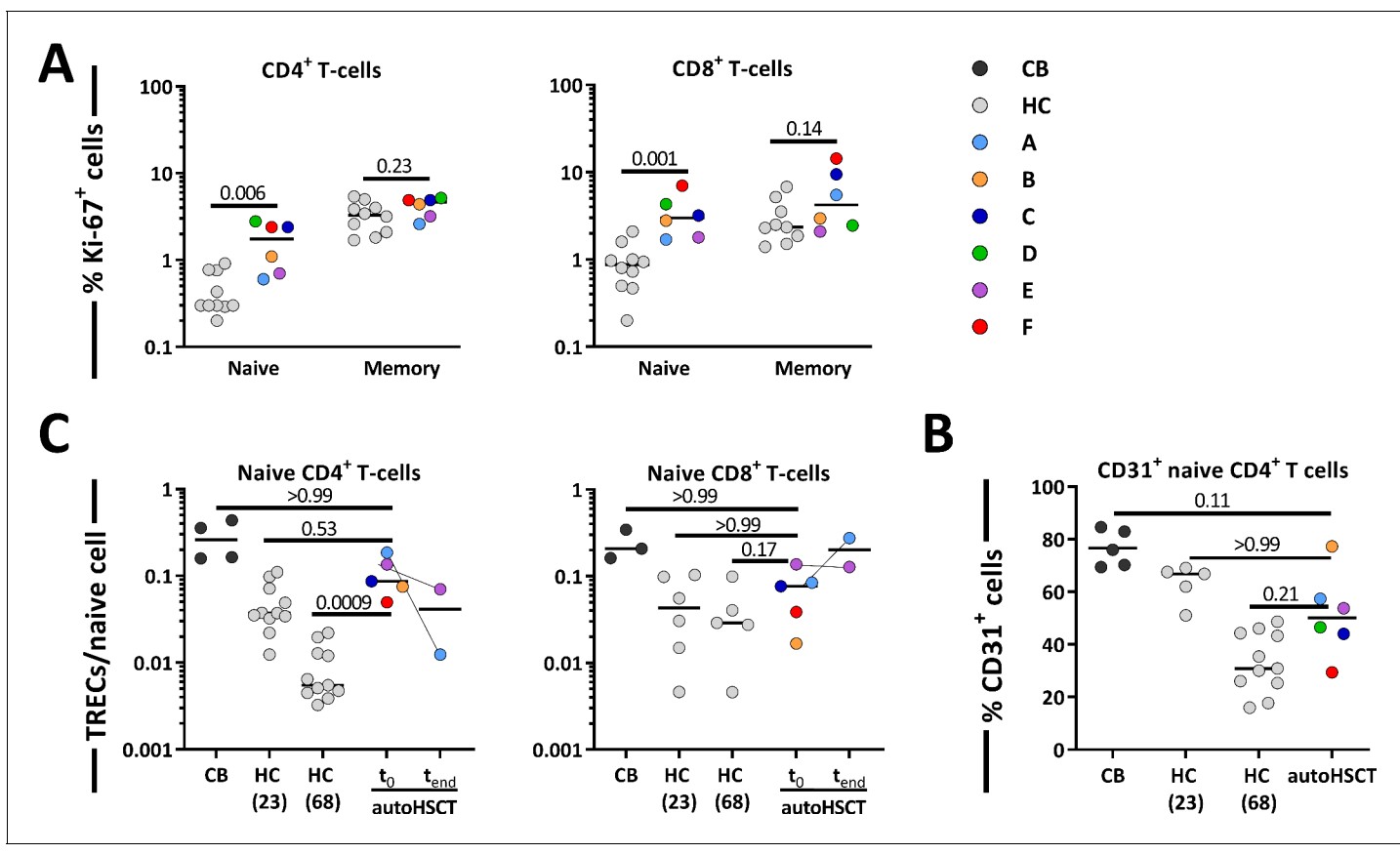

**Figure 5.** Contribution of peripheral proliferation and thymic output to T-cell production after autologous hematopoietic stem cell transplantation (autoHSCT). (A) Ki-67 expression was measured within naive and memory CD4[+] (left panel) and CD8[+] (right panel) T-cell in autoHSCT patients and healthy controls (HCs) (*Westera et al., 2015*) (for gating strategy, see *Figure 5—figure supplement 1*). (B) Average number of T-cell receptor excision circles (TRECs) per naive CD4[+] (left panel) and CD8[+] (right panel) T-cell in autoHSCT patients, cord blood (CB), and HCs (*Westera et al., 2015*). For *Patient A* and *Patient E*, TREC content was measured the first day of the study (t0) as well as the last study visit (tend). For *Patient D*, TREC content was not successfully measured due to limited material. (C) CD31 expression was measured within naive CD4[+] T-cells in autoHSCT patients, CB, and HCs (*Westera et al., 2015*). For changes in CD31 expression and absolute numbers of CD31[+] cells over time, see *Figure 5—figure supplement 2*. Different symbols indicate different individuals, autoHSCT patients (A–F) in color, CB in dark gray, and young (median age of 23 years) and old (median age of 68 years) HCs in light gray. Horizontal lines represent median values. p-values of differences between groups are shown (Mann–Whitney test [A] and Kruskal–Wallis with Dunn's correction [B], comparison with CB, HC [23], and HC [68]).

The online version of this article includes the following figure supplement(s) for figure 5:

**Figure supplement 1.** Gating strategy for Ki-67 expression of T- and B-cell subsets.

**Figure supplement 2.** CD31[+] CD4[+] T cell numbers and percentage over time.

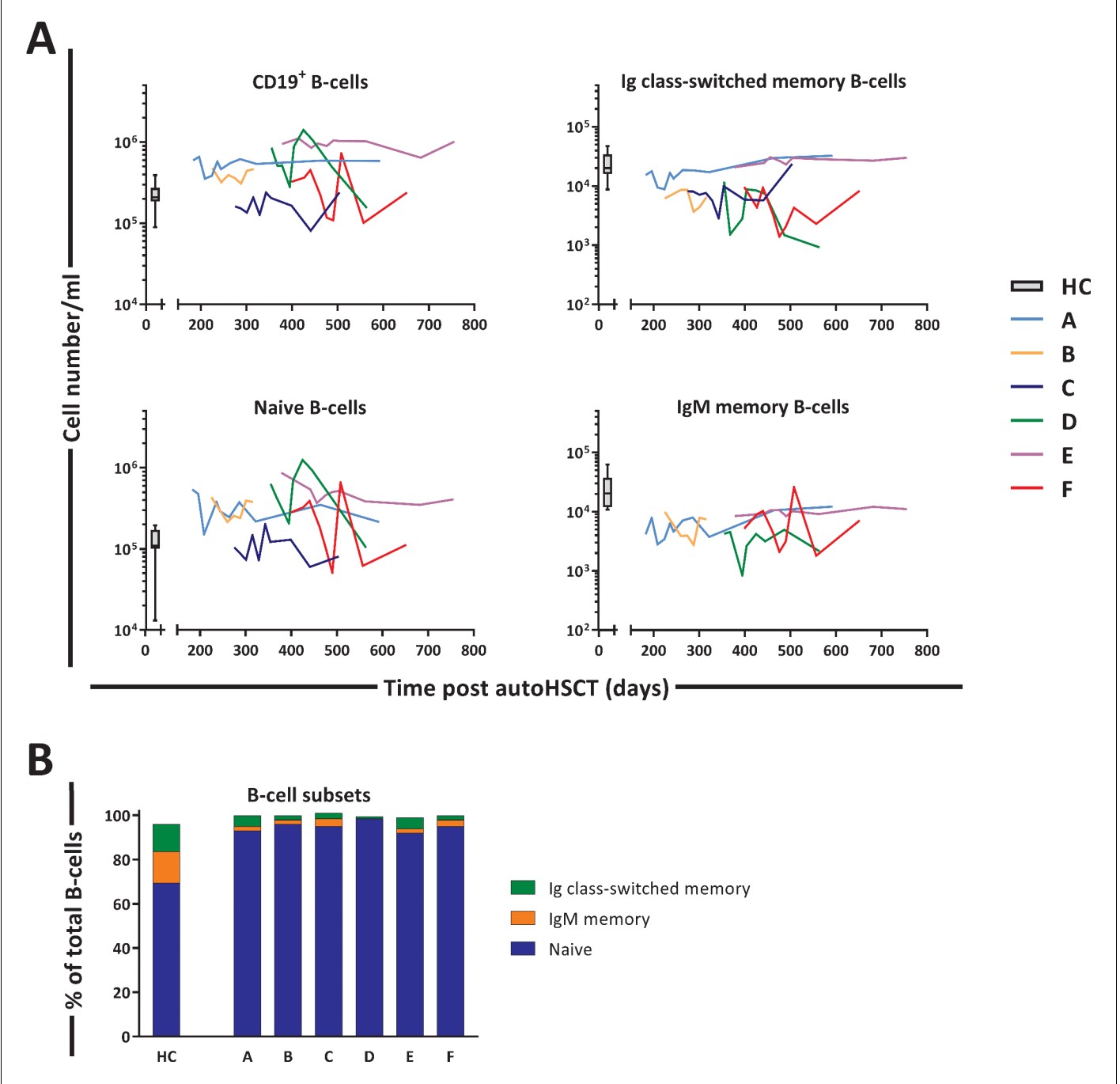

**Figure 6.** B-cell reconstitution following autologous hematopoietic stem cell transplantation (autoHSCT). (**A**) Absolute numbers (cells per milliliter) of total CD19$^+$ B-cells, naive (CD19$^+$IgM$^+$CD27$^-$), Ig class-switched memory (CD19$^+$IgM$^-$CD27$^+$), and IgM$^+$ memory (CD19$^+$IgM$^+$CD27$^+$) B-cells in peripheral blood over time. Graphs show the absolute cell counts per milliliter in autoHSCT patients (patients A–F) over the duration of the study. Box plots represent the distribution of values for healthy controls (HCs) (N = 10, box = 25th to 75th percentile, black line=median, whiskers=min and max values). Absolute numbers shown in the graph are not normalized. (**B**) Bar graphs show the median percentage of naive, Ig class-switched memory, and IgM$^+$ memory B-cells within total CD19$^+$ B-cells of autoHSCT patients (patients A–F) and HCs (N = 10). For the B-cell subset distribution per patient over time, see *Figure 6—figure supplement 1* and *Figure 6—source data 1*. Note the different y-axes in panel **A**.

The online version of this article includes the following source data and figure supplement(s) for figure 6:

**Source data 1.** B-cell numbers and percentage in blood of patients and healthy controls.

**Figure supplement 1.** B-cell subset distribution per autologous hematopoietic stem cell transplantation (autoHSCT) patient over time.

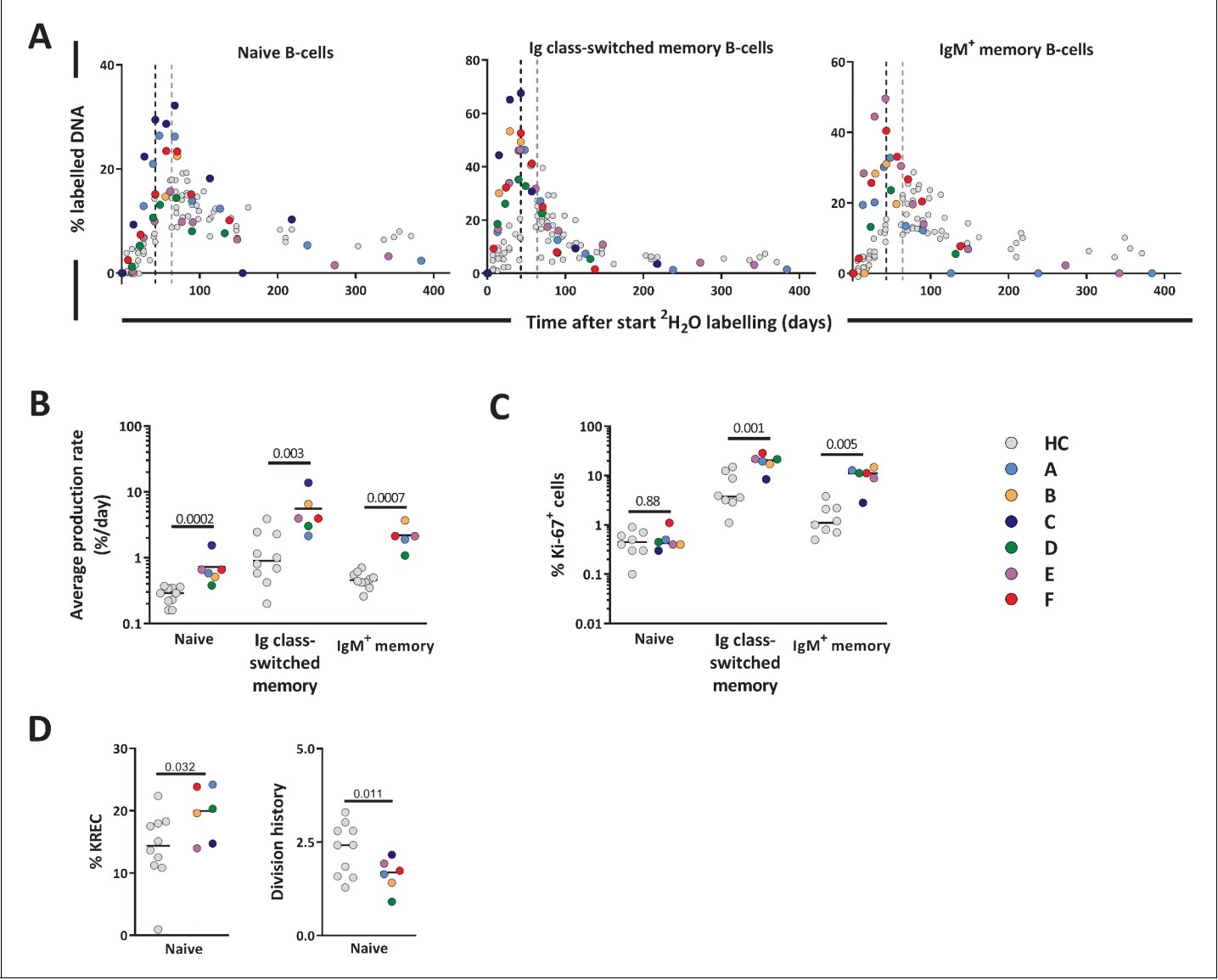

**Figure 7.** B-cell dynamics after autologous hematopoietic stem cell transplantation (autoHSCT). (A) Deuterium enrichment in the DNA of B-cell subsets in autoHSCT patients (A–F, color symbols) and healthy controls (HCs) (gray symbols) (*Westera et al., 2015*). Dotted lines correspond to the end of the labeling period (black for autoHSCT patients and gray for HCs). Label enrichment was scaled between 0% and 100% by normalizing for the maximum enrichment in granulocytes (*Figure 4—source data 1*). (B) Estimates of the per cell production rates of naive, Ig class-switched memory, and IgM$^+$ memory B-cells in autoHSCT patients and HCs (*Westera et al., 2015*). For individual fits and estimates, see *Figure 7—figure supplement 1* and *Figure 7—source data 1*. (C) Ki-67 expression was measured within naive, Ig class-switched memory, and IgM$^+$ memory B-cells in autoHSCT patients and HCs (*Westera et al., 2015*) (for gating strategy, see *Figure 5—figure supplement 1*). (D) Percentage of naive B-cells containing a KREC and naive B-cell division history for autoHSCT patients and HCs (*Westera et al., 2015*). Different symbols indicate different individuals, autoHSCT patients (A–F) in color and HCs in gray. Horizontal lines represent median values. p-values of differences between groups are shown (Mann–Whitney test). For information on modeling in R, see *Figure 4—source code 1*.

The online version of this article includes the following source data and figure supplement(s) for figure 7:

**Source data 1.** Estimates of average daily production rates for B cell subsets of autoHSCT patients.

**Figure supplement 1.** Best fits of $^2$H enrichment in B-cell subsets in autologous hematopoietic stem cell transplantation (autoHSCT) patients.

## Increased lymphocyte production rates are counteracted by increased lymphocyte loss rates

The increased lymphocyte production rates that we observed in patients after autoHSCT may at first sight suggest that, also in humans, lymphocyte production is regulated in a density-dependent

manner. The observation that lymphocyte production rates were also elevated for subsets for which cell numbers had already normalized, however, challenges this interpretation. Another observation challenging this interpretation is that for most subsets, lymphocyte numbers increased very little over time, despite the significant increase in lymphocyte production. This suggests that lymphocyte loss rates were also significantly increased after autoHSCT.

To estimate the average loss rates of all lymphocyte subsets (i.e. the number of cells lost per day, by cell death, migration, or differentiation, divided by the number of resident cells in the population), we used the average lymphocyte production rates estimated from the deuterium labeling experiments and an exponential function to describe the changes in cell numbers of each lymphocyte subset over time. For most T- and B-cell subsets, the average loss rate was approximately three to five times higher post-autoHSCT than in healthy individuals (*Figure 8* and *Figure 8—source data 1*). For naive CD8$^+$ T-cells the average loss rate was even 9.5 times higher in patients than in healthy individuals (*Figure 8* and *Figure 8—source data 1*). Thus, despite the fact that production rates are clearly increased in patients post-autoHSCT, this increased production goes hand in hand with increased lymphocyte loss rates, thereby challenging the view that it reflects a homeostatic response to low lymphocyte numbers.

## Discussion

From a homeostatic viewpoint, a response to low lymphocyte numbers could take the form of increased lymphocyte production or decreased lymphocyte loss. Based on the observation that severe lymphopenia in mice is associated with increased peripheral proliferation (*Freitas and Rocha, 2000*; *Miller and Stutman, 1984*; *Bell et al., 1987*; *Fry et al., 2001*), it is widely believed that lymphocyte production rates are increased when cell numbers are low, and normalize when cell numbers do. We have previously shown that naive T-cell production rates do not increase to compensate for the at least 10-fold declined thymic output in elderly individuals (*Westera et al., 2015*). This could be due to the relatively small degree of naive T-cell loss observed during healthy aging. Under more severe conditions of lymphopenia in humans, high frequencies of proliferating lymphocytes have been observed, but these have been linked to immune activation and clinical events, for example, GVHD and opportunistic infections (*Hazenberg et al., 2002*). Thus, there is little evidence that lymphocyte numbers regulate cell production and loss rates in humans.

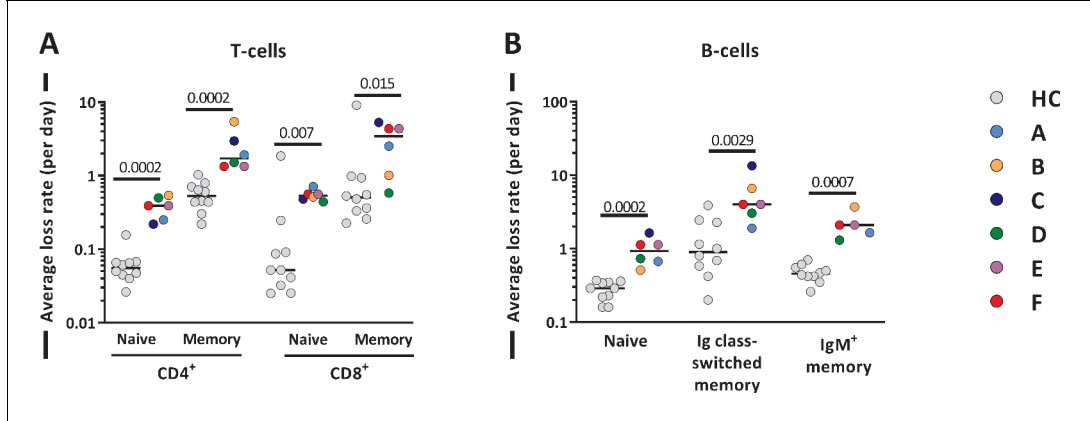

**Figure 8.** Average T- and B-cell loss rates following autologous hematopoietic stem cell transplantation (autoHSCT). (**A**) Estimates of the average loss rates of naive and memory CD4$^+$ and CD8$^+$ T-cells and of (**B**) naive, Ig class-switched memory and IgM$^+$ memory B-cells in autoHSCT patients (A–F, color symbols) and healthy controls (HCs; gray symbols) (*Westera et al., 2015*). Average loss rates were calculated (see *Figure 8—source data 1*) using the estimated average production rates and the corrected cell numbers (*Figure 8—figure supplements 1* and *2*) as described in Materials and methods. Horizontal lines represent median values. p-values of differences between groups are shown (Mann–Whitney test).

The online version of this article includes the following source data and figure supplement(s) for figure 8:

**Source data 1.** Estimates of average daily loss rates for T- and B-cell subsets autoHSCT patients.

**Figure supplement 1.** Best fits of T-cell numbers in autologous hematopoietic stem cell transplantation (autoHSCT) patients.

**Figure supplement 2.** Best fits of B-cell numbers in autologous hematopoietic stem cell transplantation (autoHSCT) patients.

Our deuterium labeling study shows that in patients receiving an autoHSCT, in the absence of GVHD, clinically manifested infections, and transplantation-related complications, the production rates of most T- and B-cell subsets were significantly increased 12 months after transplantation. Increased lymphocyte production rates during lymphopenia have generally been interpreted as evidence for a density-dependent response to low lymphocyte numbers (*Jones et al., 2013*; *Bouvy et al., 2013*; *Alho et al., 2016*; *Havenith et al., 2012*; *Bouvy et al., 2016*). We did two additional observations, however, that suggested that these increased lymphocyte production rates post-autoHSCT were not simply reflecting a homeostatic response to low cell numbers. First, T- and B-cell production rates did not normalize when cell numbers did. Second, not only lymphocyte production but also lymphocyte loss rates were significantly increased post-autoHSCT. Alternatively, the observed increased lymphocyte production rates post-autoHSCT could reflect an overrepresentation of young lymphocytes within the lymphocyte pool, in analogy with lymphocytes in the developing immune system of children, where especially in the first year T-cells have relatively high proliferation rates (*van Gent et al., 2009*). Likewise, in mice it has been shown that newly generated naive and memory T-cells have higher production and loss rates than their established counterparts (*Gossel et al., 2017*; *Rane et al., 2018*; *Reynaldi et al., 2019*). Finally, despite the fact that the patients in our study were included up to 12 months post-transplantation and were selected on the basis of being in good health, we cannot exclude the possibility that the increased proliferation and loss rates observed post-autoHSCT may reflect HSCT-related complications, such as the impact of initial chemotherapy and conditioning-therapy or subclinical infections and inflammation, which may have gone unnoticed. In fact, the increased lymphocyte production rates may even have been a response to increased lymphocyte loss rates, which themselves may have been induced by the transplantation. Lymphocyte production would then be modulated in an effort to normalize lymphocyte numbers. Whatever the underlying explanation, the ongoing dysregulation of lymphocyte dynamics in itself is an important observation. It shows that normalized cell numbers cannot be taken as an indication that homeostasis has been restored.

The observation that lymphocyte loss rates were increased post-autoHSCT is quite remarkable in light of the widely held view that homeostatic mechanisms could take the form of increased lymphocyte survival. This concept is supported by the observation that the availability of pro-survival and anti-apoptotic factors, such as IL-7 (*Ponchel et al., 2011*; *Barata et al., 2019*; *Lundström et al., 2012*), typically increases during lymphopenia. In our cohort, we found no evidence for increased IL-7 plasma levels, consistent with the observation that lymphocyte loss rates in these patients were not decreased. It remains unclear why IL-7 plasma levels were not increased in our cohort. A possible explanation could be that IL-7 production was hampered by transplantation-related damage to, for example, stromal cells or intestinal epithelial cells, which are an important source of IL-7 (*Barata et al., 2019*; *Lundström et al., 2012*; *Kim et al., 2011*). We found that lymphocyte loss rates were up to 10-fold increased after autoHSCT. Although in the current study we did not take into account markers of cell death, the estimated increased cell loss rates are in line with previous human studies on T-cell survival after allogenic HSCT, which consistently reported that the fraction of pro-apoptotic cells increases following transplantation (*Alho et al., 2016*; *Brugnoni et al., 1999*; *Lin et al., 2000*; *Poulin et al., 2003*). Although this suggests that intervention with lymphocyte survival after HSCT may aid lymphocyte reconstitution, other factors apart from increased cell death may have contributed to the loss of cells from the peripheral blood. Excessive lymphocyte differentiation and/or increased migration to the tissues would also increase cell loss rates. Further studies should address whether lymphocyte reconstitution occurs at similar rates in blood and tissues in order to clarify whether lymphocyte recruitment to the tissues may be a key factor influencing the loss of lymphocytes from the blood following autoHSCT.

Consistent with previous reports (*Bosch et al., 2012*; *Alho et al., 2016*; *Storek et al., 2008*), we found that 12 months post-autoHSCT, CD4$^+$ T-cell numbers were below the normal range while CD8$^+$ T-cells recovered more rapidly. Deuterium labeling in patients revealed that the average production rates of most T-cell subsets were significantly increased following autoHSCT. This increase was especially evident for naive T-cells. The high percentage of Ki-67$^+$ naive T-cells post-autoHSCT suggests that increased naive T-cell production is to a large extent explained by increased peripheral T-cell proliferation. Memory T-cell production rates based on deuterium enrichment were also higher in patients compared to controls, while Ki-67 expression suggested that memory CD4$^+$ and CD8$^+$ T-cell proliferation rates were not significantly increased 0.5–1 year after autoHSCT in line with

previous reports (*Malphettes et al., 2003*). This seeming contradiction may be explained by the fact that Ki-67, a snapshot marker, may be less sensitive to detect differences in T-cell proliferation than long-term in vivo deuterium labeling. Alternatively, the increased production rate of memory T-cells post-autoHSCT may be due to increased transition of naive T-cells into the memory T-cell population. In line with this, in mice it has been demonstrated that naive T-cells adoptively transferred into immunodeficient animals can acquire a memory phenotype after antigen independent stimulation and division (*Cho et al., 2000*; *Goldrath et al., 2000*).

If a significant part of cell production in a certain lymphocyte subset (e.g. the memory subset) is indeed due to an influx from another lymphocyte subset (e.g. the naive subset), the increased production rates that we observed may reflect either a true increase in cell production or a normal influx of cells entering a smaller lymphocyte population. To distinguish between these options, for each lymphocyte subset and each individual, we also calculated the total number of cells produced per day (i.e. coming from a source and/or from peripheral cell division), by multiplying the average production rate of each lymphocyte subset with the median cell number of that subset, and compared these values to those in HCs (*Westera et al., 2015*; *Figure 3—source data 1* and *Figure 4—source data 2*). We found that total daily lymphocyte production was as high as in HCs for naive CD4$^+$ T-cells and higher than in HCs for all other lymphocyte subsets, suggesting that the increased lymphocyte production rates post-autoHSCT truly reflected increased T-cell proliferation and/or an increased influx from another lymphocyte compartment.

Measuring thymopoiesis and the contribution of RTEs to the naive T-cell pool after HSCT is not straightforward. Although increased TREC contents at first sight seem suggestive for increased thymic output, T-cells bearing TRECs may in fact be overrepresented in the peripheral T-cell pool post-transplantation when cell numbers are low (*Hazenberg et al., 2003*). Hence, for naive CD4$^+$ T-cells, whose numbers had not yet normalized, increased average TREC contents may incorrectly be interpreted as evidence for increased thymic output. The finding that the average TREC content of naive CD4$^+$ T-cells following autoHSCT was higher than in age-matched controls provides no evidence that thymic output following transplantation was higher than in HCs, but does imply that the thymus had become functional again within 12 months after intense conditioning for autoHSCT. The fact that the average TREC content of naive CD4$^+$ T -cells, but not that of naive CD8$^+$ T-cells, was higher in patients than in healthy individuals may reflect differences in the degree of depletion of naive CD4$^+$ and CD8$^+$ T -cells. Alternatively, it might reflect differences in the way CD4$^+$ and CD8$^+$ T-cells are generated. In support of the latter explanation, repertoire analyses in patients receiving an autoHSCT for the treatment of autoimmune diseases have suggested that CD4$^+$ T-cells largely arise de novo, since most CD4$^+$ T-cell clones post-autoHSCT were not present at baseline, while CD8$^+$ T-cells mainly expand from cells that were already circulating pre-transplantation (*Muraro et al., 2005*; *Muraro et al., 2014*; *Dubinsky et al., 2010*).

To study in a population other than T-cells whether lymphocyte production and loss rates in humans are regulated in a density-dependent manner, we quantified the production and loss rates of different B-cell subsets. In line with previous reports (*Burns et al., 2003*; *Avanzini et al., 2005*; *Bemark et al., 2012*), we found that 12 months after transplantation, naive B-cell numbers had reconstituted to healthy (or even higher than healthy) control values, while Ig class-switched and IgM$^+$ memory B-cells had not yet fully recovered. The delayed reconstitution of Ig class-switched and IgM$^+$ memory B-cells has typically been attributed to treatment-related damage to secondary lymphoid organs, which may hamper the formation of germinal centers essential for somatic hypermutation and isotype switching (*Avanzini et al., 2005*). Also for naive, Ig class-switched, and IgM$^+$ memory B-cells, we found that not only production rates but also cell loss rates were increased 12 months post-autoHSCT, further supporting our conclusion that increased lymphocyte production rates do not simply reflect a homeostatic response to low lymphocyte numbers.

In brief, our findings show that despite the slow reconstitution of lymphocytes in autoHSCT patients, lymphocyte production rates are increased. Since this increased production goes hand in hand with increased cell loss and does not normalize when cell numbers do, it is not simply due to a homeostatic response to low cell numbers. Future studies should address whether the dynamics of lymphocytes after autoHSCT normalize in the long run, what drives the increase in lymphocyte production and loss rates during immune reconstitution, and to what extent immune reconstitution in the tissues occurs.

# Materials and methods

## Key resources table

| Reagent type (species) or resource | Designation | Source or reference | Identifiers | Additional information |
|---|---|---|---|---|
| Antibody | Anti-human CD45-PerCP (Mouse IgG1, κ) RRID:AB_2566358 | BioLegend | Cat# 368506 Clone: 2D1 | '(1:20)' |
| Antibody | Anti- human CD3-FITC (Mouse IgG1, κ) RRID:AB_2562046 | BioLegend | Cat# 399430 Clone: UCHT1 | '(1:25)' |
| Antibody | Anti-human CD4-APC-eF780 (Mouse IgG1, κ) RRID:AB_1272044 | eBioscience | Cat# 47-0049-42 Clone: RPA-T4 | '(1:50)' |
| Antibody | Anti-human CD8-V500 (Mouse IgG1, κ) RRID:AB_2870326 | BD Biosciences | Cat# 561617 Clone: SK1 | '(1:60)' |
| Antibody | Anti-human CD19-eFluor450 (Mouse IgG1, κ) RRID:AB_1272053 | eBioscience | Cat# 48-0199-42 Clone: HIB19 | '(1:25)' |
| Antibody | Anti-human CD45RO-PE-Cy7 (Mouse IgG2A, κ) RRID:AB_647426 | BD Biosciences | Cat# 337168 Clone: UCHL1 | '(1:60)' |
| Antibody | Anti-human CD27-APC (Mouse IgG1, κ) RRID:AB_469371 | eBioscience | Cat# 17-0279-42 Clone: O343 | '(1:25)' |
| Antibody | Anti-human CD31-PE (Mouse IgG1, κ) RRID:AB_400016 | BD Biosciences | Cat# 340297 Clone: L133.1 | '(1:12.5)' |
| Antibody | Anti-human CD3-eFluor450 (Mouse IgG2A, κ) RRID:AB_1272055 | eBioscience | Cat# 48-0037-42 Clone: OKT3 | '(1:50)' |
| Antibody | Anti-human CD95-PE (Mouse IgG1, κ) RRID:AB_396027 | BD Biosciences | Cat# 555674 Clone: DX2 | '(1:50)' |
| Antibody | Anti-human CD19-PerCP (Mouse IgG1, κ) RRID:AB_2868816 | BD Biosciences | Cat# 363014 Clone:SJ25C1 | '(1:50)' |
| Antibody | Anti-human IgM-PE (Goat IgG) RRID:AB_2795614 | Southern Biotech | Cat# 2022–09 Polyclonal | '(1:100)' |
| Antibody | Anti-human Ki-67-FITC (Mouse IgG1, κ) RRID:AB_578716 | DAKO | Cat# F7268 Clone:MIB-1 | '(1:10)' |
| Commercial assay or kit | Cytofix/Cytoperm | BD Biosciences | Cat#554714 | |
| Commercial assay or kit | FACS Lysing Solution | BD Biosciences | Cat#349202 | |
| Commercial assay or kit | Reliaprep Blood gDNA Miniprep System | Promega | Cat#A5081 | |
| Commercial assay or kit | Data acquisition | Luminex | xPONENT software version 4.2 | |
| Commercial assay or kit | Data acquisition | Biorad Laboratories | Biorad FlexMAP3D | |
| Chemical compound | $^2H_2O$, 99.8% enriched | Cambridge Isotope Laboratories | Cat#DLM-2259–1 | |

*Continued on next page*

*Continued*

| Reagent type (species) or resource | Designation | Source or reference | Identifiers | Additional information |
|---|---|---|---|---|
| Software, algorithm | Data analysis | Biorad laboratories | Biorad Bio-Plex Manager software, version 6.1.1 | |
| Software, algorithm | Data analysis | Biorad laboratories | Biorad FlexMAP3D | |
| Software, algorithm | Data analysis | *Westera et al., 2013* DOI: [10.1182/blood-2013-03-488411](10.1182/blood-2013-03-488411) | Multiexponential model | |
| Software, algorithm | Data analysis | GraphPad PRISM | GraphPad Software | |

## Patient characteristics

Six patients who received an autoHSCT for the treatment of a hematologic malignancy were enrolled in the study after having provided written informed consent. Following repeated subcutaneous injections with granulocyte-colony stimulating factor (G-CSF), stem cells were obtained by leukapheresis of peripheral blood. Patients received a non T-cell depleted graft; the average number of $CD34^+$ cells transplanted was $5.03 \times 10^6$ cells/kg (median, 4.12; range, 1.82–12.38). Patients were included in the study between 196 and 420 days after autoHSCT, and had no signs of transplantation-related complications, severe infections (HIV, HBV, and HCV), other liver diseases, active uncontrolled infections (such as infectious mononucleosis), inadequate liver or kidney function, or cardiovascular disease before and during the study. Additional inclusion criteria were: fully transfusion independent at start of the study, hemoglobin level $\geq$6 mmol/l, and platelet count $\geq 50 \times 10^9$/L. Any use of medication during the study was unrelated to the malignancy and the HSCT (*Figure 1*). In order to compare the phenotypes of the B- and T-cell compartments of patients to those of age-matched healthy individuals, we used data from healthy individuals from a previous study (*Westera et al., 2015*), additional blood samples were collected from healthy volunteers not following the labeling protocol after having provided informed consent. This study was approved by the medical ethical committee of the University Medical Center Utrecht and conducted in accordance with the Helsinki Declaration.

## In vivo deuterium labeling

In vivo deuterium labeling was performed as previously described with small adaptations (*Westera et al., 2015*). Briefly, patients received an oral ramp-up dose of 7.5 ml of heavy water ($^2H_2O$, 99.8% enriched, Cambridge Isotope Laboratories) per kilogram body water on the first day of the study, and drank a daily maintenance dose of 1.25 ml $^2H_2O$ per kilogram body water for 6 weeks. To reduce the study burden, the labeling period of autoHSCT patients was 3 weeks shorter than the labeling period we used in our previous study in HCs (*Westera et al., 2015*). Thanks to the use of the multi-exponential model, these different labeling periods for autoHSCT patients and HCs should not affect the estimated dynamic parameters (*Westera et al., 2013*). Blood was withdrawn four times during the labeling period and six times during the de-labeling period, with the last withdrawal approximately 1 year after the start of $^2H_2O$ administration. Urine samples were collected during the first 13 weeks of the study and stored at $-20°C$ until analysis. For deuterium labeling, data from HCs from a previous study (*Westera et al., 2015*) were used. The age of the autoHSCT patients included in the current study (median age of 54 years, see *Figure 1B*) was not completely matched with that of the HCs studied before (median age of 22 years for young and 68 years for aged controls). Since lymphocyte dynamics hardly change with age (*Westera et al., 2015*), the comparisons in our study should not be affected by the relatively small age differences between patients and HCs.

## Cell isolation, flow cytometry, and cell sorting

Peripheral blood mononuclear cells were obtained by Ficoll-Paque (GE Healthcare, Little Chalfont, UK) density gradient centrifugation from heparinized blood. Granulocytes were obtained by two cycles of erythrocyte lysis (155 mM $NH_4Cl$, 10 mM $KHCO_3$, 0.1 mM $Na_2$-EDTA, pH = 7.0) of the granulocyte/erythrocyte layer. To determine the baseline deuterium enrichment, total peripheral blood mononuclear cells were frozen on the first day of the study, prior to $^2H_2O$ intake.

Absolute cell numbers were determined using TruCount tubes (BD Biosciences, San Jose, CA, USA), in which whole blood was stained using CD45-PerCP (BioLegend), CD3-FITC (BioLegend),

CD4-APC-eF780 (eBioscience), CD8-V500 (BD Biosciences), CD19-eFluor450 (eBioscience), CD45RO-PE-Cy7 (BD Biosciences), CD27-APC (eBioscience), and CD31-PE (BD Biosciences) antibodies. After erythrocyte lysis with FACS Lysing Solution (BD Biosciences), samples were immediately analyzed.

For cell cycle analysis, we analyzed the expression of the nuclear protein Ki-67. Cells were first extracellularly stained with CD3-eFluor450 (eBioscience), CD4 APC-eFluor780 (eBioscience), CD8-V500 (BD Biosciences), CD45RO-PE-Cy7 (BD), CD27-APC (eBioscience), and CD95-PE (BD Biosciences) or with CD19-PerCP (Biolegend), CD27-APC (eBioscience), and IgM-PE (Southern Biotech) monoclonal antibodies. Subsequently, cells were fixed and permeabilized (Cytofix/Cytoperm, BD Biosciences) and stained intracellularly with Ki-67-FITC (DAKO, Glostrup). Washing steps were carried out using Perm/Wash buffer (BD Biosciences). Samples were analyzed on an LSR-II or LSR-Fortessa flow cytometer using FACS Diva software (BD Biosciences).

For sorting of B- and T-cell subsets, cells were incubated with CD3-eFluor450 (eBioscience), CD4 APC-eFluor780 (eBioscience), CD8-PE (Biolegend), CD19-PerCP (Biolegend), CD45RO-PE-Cy7 (BD), CD27-APC (eBioscience), and IgM-FITC (Southern Biotech). CD19$^+$ naive (IgM$^+$CD27$^-$), Ig class-switched (IgM$^-$CD27$^+$), and IgM$^+$ (IgM$^+$CD27$^+$) memory B-cells and CD3$^+$CD4$^+$ and CD3$^+$CD8$^+$ naive (CD27$^+$CD45RO$^-$) and memory (CD45RO$^+$) T-cells were sorted on a FACSAria II or FACSAria III cell sorter using FACS Diva software (BD Biosciences). Flow cytometric analyses and cell sorting were always performed on freshly isolated material. Representative density dot plots and the gating strategy for TruCount and cell sorting are shown in *Figure 3—figure supplement 1*.

## DNA isolation
Genomic DNA was isolated from sorted B- and T-cell subsets, total peripheral blood mononuclear cells, and granulocytes using the Reliaprep Blood gDNA Miniprep System (Promega, Madison, WI, USA) and stored at $-20°C$ before processing for TREC analysis or gas chromatography/mass spectrometry (GC/MS).

## TREC analysis
In sorted naive CD4$^+$ and CD8$^+$ T-cell samples, signal joint TREC numbers and DNA input were quantified with a ViiA 7 Real-Time PCR System (Applied Biosystems) as previously described (*Hazenberg et al., 2000*; *Goldrath et al., 2000*). Values are the mean of two qPCR measurements of a given DNA sample.

## KREC analysis
For quantification of B-cell replication history, the KREC assay in sorted naive B-cells was performed as described previously (*Muraro et al., 2005*). In short, genomic DNA was used as template for Taq-Man-based real-time quantitative PCR of the albumin control gene, intronRSS–Kde coding joints from rearranged IGK loci, and intronRSS–Kde signal joints on KRECs. The difference in Ct values between the intronRSS–Kde coding joints and signal joints from the same sample was used to calculate B-cell replication history with technical correction Ct values obtained with the U698-DB01 control cell line (*Muraro et al., 2005*) as follows (*Muraro et al., 2014*):

$$\left(\mathrm{Ct_{signal\,joint}} - \mathrm{Ct_{coding\,joint}}\right)_{\mathrm{sample}} - \left(\mathrm{Ct_{signal\,joint}} - \mathrm{Ct_{coding\,joint}}\right)_{\mathrm{U698-DB01}}$$

The frequencies of cells containing an intronRSS–Kde coding joint were calculated as follows (*Muraro et al., 2014*):

$$2^{\left[\left(\mathrm{Ct_{albumin}} - \mathrm{Ct_{coding\,joint}}\right)_{\mathrm{sample}} - \left(\mathrm{Ct_{albumin}} - \mathrm{Ct_{coding\,joint}}\right)_{\mathrm{U698-DB01}}\right]} \times 100\%$$

## Multiplex immunoassay
Plasma samples were obtained from heparinized blood at different time points and stored at $-80°C$. Plasma levels of CRP (C-reactive protein), APRIL (A proliferation-inducing ligand or TNFSF13), FAS-L (FAS ligand or CD95L), IL-7 (Interleukin 7), IL-15 (Interleukin 15), and IL-8 (Interleukin eight or CXCL8) were measured by a multiplex immunoassay using Luminex xMAP technology (xMAP, Luminex, Austin, TX, USA). The assay was performed as previously described (*Dubinsky et al., 2010*). Biorad FlexMAP3D (Biorad laboratories, Hercules, USA) and xPONENT software version 4.2

(Luminex) were used for acquisition and data was analyzed by five-parametric curve fitting using Bio-Plex Manager software, version 6.1.1 (Biorad). Samples were measured without any previous freeze-thaw cycle. Samples of HCs were acquired at the same days as patient samples to take into account the effect of storage on the different plasma markers.

## Measurement of deuterium enrichment in DNA and body water

Deuterium enrichment in DNA from granulocytes, sorted cells, and total peripheral blood mononuclear cells (t=0) was measured according to the method described by *Burns et al., 2003*. with minor modifications (*Westera et al., 2015*). Briefly, DNA was enzymatically hydrolyzed into deoxyribonucleotides and derivatized to penta-fluoro-triacetate (PFTA) before injection (DB-17MS column, Agilent Technologies) into the gas chromatograph (7890A GC System, Agilent Technologies). PFTA was analyzed by negative chemical ionization mass spectrometry (5975C inert XL EI/CI MSD with Triple-Axis Detector, Agilent Technologies) measuring ions m/z 435 and m/z 436. For quantification of 2H enrichment, standard solutions with known enrichment (Tracer-to-Tracee ratios ([M + 1]/[M + 0]) 0, 0.0016, 0.0032, 0.0065, 0.0131, 0.0265, 0.0543, and 0.1140) were made by mixing 1-13C-deoxyadenosine (Cambridge Isotopes Inc; generates an 'M + 1' ion) with unlabeled deoxyadenosine (Sigma, St. Louis, MO, USA). To correct for abundance sensitivity of isotope ratios, we followed the approach proposed by Patterson et al (*Avanzini et al., 2005*) on log 10-transformed enrichment data. Deuterium enrichment in urine was analyzed on the same GC/MS system (using a PoraPLOT Q 25 × 0.32 column, Varian) by electron impact ionization as previously described (*Bemark et al., 2012*). Values are the mean of two GCMS measurements of a given derivative sample.

## Quantification of lymphocyte dynamics by mathematical modeling of urine and DNA enrichment data and cell numbers

Mathematical models were fitted to the urine and DNA enrichment data as previously described (*van Gent et al., 2011*). The estimated maximum level of $^2$H enrichment in the granulocyte population of each patient was considered to be the maximum level of label incorporation that cells could possibly attain and was used to scale the enrichment data of the other cell subsets. As cell numbers may not be constant over time during the lymphocyte reconstitution phase, we adapted the commonly used mathematical model for deuterium labeling in T- and B-cells by releasing the steady-state assumption. For naive T- and B-cells ($N$), we allow cells to be produced by the thymus for T-cells and the bone marrow for B-cells at rate $\sigma$, proliferate at a rate $p_N$, and are lost at a rate $d_N$. For the other lymphocyte subsets ($X$), we write that they proliferate at a rate $p_X$ and are lost at a rate $d_X$.

$$\frac{dN}{dt} = \sigma + p_N N - d_N N \tag{1}$$

$$\frac{dX}{dt} = p_X X - d_X X \tag{2}$$

The total amount of labeled DNA ($L$) can be modeled by the following differential equations:

$$\frac{dL_N}{dt} = \sigma\, cU(t) + p_N\, cU(t)\, N - d_N L_N \tag{3}$$

$$\frac{dL_X}{dt} = p_X\, cU(t)\, X - d_X L_X \tag{4}$$

where $U(t)$ is the fraction of deuterium in body water at time $t$ (in days), and $c$ is an amplification factor as previously described (*van Gent et al., 2011*; *Westera et al., 2013*). We can derive the equations for the fraction of labeled DNA (defined by $l = L/X$) using the quotient rule of differentiation:

$$\frac{dl_N}{dt} = \left(\frac{\sigma}{N} + p_N\right)(cU(t) - l_N) \tag{5}$$

$$\frac{dl_X}{dt} = p_X \left( cU(t) - l_X \right) \tag{6}$$

Because of parameter identifiability issues, we were not able to estimate both $\sigma$ and $p_N$ in *Equation 5*. We therefore made the simplifying assumption that the per cell production rate (i.e. the number of new cells produced per day, coming from the source or peripheral cell division, divided by the number of resident cells in the population) $\left( \frac{\sigma}{N} + p_N \right)$ was not time-dependent, such that *Equation 6* could also be used for naive T- and B-cells. To account for potential kinetic heterogeneity in each subpopulation (*Westera et al., 2013*), we used a multi-exponential model in which each subpopulation $i$ contains a fraction $\alpha_i$ of cells with production rate $p_i$ per day, and made the simplifying assumption that these fractions $\alpha_i$ are not time-dependent. The average per cell production rate $p$ of each subpopulation was subsequently calculated as $p_X = \sum_i \alpha_i p_i$ (*Vrisekoop et al., 2008*; *Ganusov et al., 2010*). All subsets, except naive CD4$^+$ and CD8$^+$ T-cells, were significantly better described by two subpopulations. For naive CD4$^+$ and CD8$^+$ T-cells, only one was needed.

*Equation 6* shows that deuterium enrichment data will give information only on the per cell proliferation rates $p_X$. To estimate the loss rates of the different lymphocyte subsets (i.e. the number of cells lost per day, by cell death, migration or differentiation, divided by the number of resident cells in the population) $(d_X)$ we made use of cell number data. As leukocyte counts in blood are known to vary, for example, due to diurnal rhythms, we first used a linear regression model to describe the total leukocyte numbers for each individual. To obtain cell numbers, the number of leukocytes according to this regression line was multiplied by the fraction of cells in each subset. We then fitted an exponential function to these 'normalized' cell numbers for each subset and individual:

$$X(t) = X_0 e^{(p_X - d_X)t} \tag{7}$$

where $X_0$ is the cell number at the time of inclusion in the study and $p_X$ was fixed to the estimated values from the deuterium analyses.

Best fits of deuterium enrichment in body water and granulocytes are shown in *Figure 4—figure supplement 1* and the corresponding parameter estimates are given in *Figure 4—source data 1*. Individual enrichment data and best fits are shown in *Figure 4—figure supplement 2* for T-cells and *Figure 7—figure supplement 1* for B-cells and the corresponding parameter estimates are given in *Figure 4—source data 2* (T-cell subsets) and *Figure 7—source data 1* (B-cell subsets). Normalized cell numbers and best fits are shown in *Figure 8—figure supplement 1* for T-cells and *Figure 8—figure supplement 2* for B-cells and the corresponding parameter estimates are given in *Figure 8—source data 1*.

## Statistical analyses

For each individual urine and granulocyte enrichment levels were simultaneously used to estimate their respective parameters and for each lymphocyte subset deuterium enrichment data and normalized cell numbers were simultaneously used to estimate their production and loss rates. Deuterium enrichment data were arcsin-sqrt transformed and normalized cell numbers were log10-transformed before parameter estimation. 95% confidence limits were determined by bootstrap analysis on the residuals. Parameter estimation was performed using a maximum likelihood approach using R (*Sauce et al., 2012*). Estimated medians of enrichment data, median values of longitudinal data, or values of single measurements were compared between two groups using Mann–Whitney tests (GraphPad Software, Inc) or multiple groups using Kruskal–Wallis with Dunn's correction. Differences with a p-value <0.05 were considered significant.

## Acknowledgements

We thank the patients for their participation in this study, Jeroen F van Velzen, Pien AJ van der Burght, and Gerrit Spierenburg for assistance with cell sorting, Laura Ackermans for theoretical input, Lyanne Derksen for critically reading the manuscript, Mr Benjamin Bartol and Ms Pei Mun Aui for technical support, and the nurses from the Julius Center Trial Unit for taking care of the study participants.

## Additional information

### Competing interests

Jürgen Kuball: reports grants from Novartis, Miltenyi Biotech, and Gadeta. Is inventor on multiple patents dealing with γδ T-cell research, ligands, and isolation techniques, and is scientific co-founder and shareholder of Gadeta. (Patent number: 9546998, 9891211, 10324083, 10578609. Publication number: 20200368278, 20200363397, 20190271688, 2019020961, 201901692603, 20180188234, 20170319674, 20170174741, 20150050670). The other authors declare that no competing interests exist.

### Funding

| Funder | Grant reference number | Author |
|---|---|---|
| European Union Seventh Framework Programme | FP7-PEOPLE-2012-ITN 317040-QuanTI | Mariona Baliu-Piqué |
| Landsteiner Foundation for Blood Transfusion Research | LSBR grant 0812 | Vera van Hoeven |

The funders had no role in study design, data collection and interpretation, or the decision to submit the work for publication.

### Author contributions

Mariona Baliu-Piqué, Vera van Hoeven, Conceptualization, Data curation, Formal analysis, Investigation, Methodology, Writing - original draft, Writing - review and editing; Julia Drylewicz, Conceptualization, Formal analysis, Investigation, Writing - original draft, Writing - review and editing; Lotte E van der Wagen, Anke Janssen, Sigrid A Otto, Investigation, Methodology, Writing - review and editing; Menno C van Zelm, Formal analysis, Supervision, Investigation, Writing - review and editing; Rob J de Boer, Conceptualization, Formal analysis, Supervision, Writing - original draft, Writing - review and editing; Jürgen Kuball, Conceptualization, Supervision, Writing - review and editing; Jose AM Borghans, Conceptualization, Formal analysis, Supervision, Funding acquisition, Investigation, Methodology, Writing - original draft, Project administration, Writing - review and editing; Kiki Tesselaar, Conceptualization, Data curation, Formal analysis, Supervision, Funding acquisition, Investigation, Methodology, Writing - original draft, Project administration, Writing - review and editing

### Author ORCIDs

Mariona Baliu-Piqué [ID] https://orcid.org/0000-0002-9276-8839
Julia Drylewicz [ID] https://orcid.org/0000-0002-9434-8459
Rob J de Boer [ID] http://orcid.org/0000-0002-2130-691X
Kiki Tesselaar [ID] https://orcid.org/0000-0002-9847-0814

### Ethics

Human subjects: This study was approved by the medical ethical committee of the University Medical Center Utrecht and conducted in accordance with the Helsinki Declaration. Six patients who received an autoHSCT for the treatment of a hematologic malignancy were enrolled in the study after having provided written informed consent.

### Decision letter and Author response

Decision letter https://doi.org/10.7554/eLife.59775.sa1
Author response https://doi.org/10.7554/eLife.59775.sa2

## Additional files

### Supplementary files

• Transparent reporting form

## Data availability

All data analysed during this study are included in the manuscript. Source data is added as separate files for Figure 2, 3, 4, 6,7 and 8.

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
