## [Decision Letter]

**Acceptance summary:**

This paper will be of interest to a broad range of immunologists and is also highly relevant to clinicians. It is a rigorous and thorough study of both T and B lymphocyte dynamics in humans, and challenges the current understanding of lymphocyte homeostasis. The key conclusions of the manuscript are well supported by the data and the performed analyses.

**Decision letter after peer review:**

Thank you for submitting your article "Cell-density independent increased lymphocyte production and loss rates post-autologous HSCT" for consideration by *eLife*. Your article has been reviewed by three peer reviewers, and the evaluation has been overseen by a Reviewing Editor and Aleksandra Walczak as the Senior Editor. The following individuals involved in review of your submission have agreed to reveal their identity: Andrew J Yates (Reviewer #2); Ken Duffy (Reviewer #3).

The reviewers have discussed the reviews with one another and the Reviewing Editor has drafted this decision to help you prepare a revised submission.

As the editors have judged that your manuscript is of interest, but as described below that additional work is required before it is published, we would like to draw your attention to changes in our revision policy that we have made in response to COVID-19 (https://elifesciences.org/articles/57162). First, because many researchers have temporarily lost access to the labs, we will give authors as much time as they need to submit revised manuscripts. We are also offering, if you choose, to post the manuscript to bioRxiv (if it is not already there) along with this decision letter and a formal designation that the manuscript is "in revision at *eLife*". Please let us know if you would like to pursue this option. (If your work is more suitable for medRxiv, you will need to post the preprint yourself, as the mechanisms for us to do so are still in development.)

Summary:

In this paper, the authors have carried out a comprehensive analysis of lymphocyte dynamics in humans after autologous hematopoietic stem cell transplant (HSCT). Using data from six week long deuterium labeling, their main finding is that rates of production and loss of all T and B subsets analyzed are elevated in auto-HCT patients, even after their numbers appear to have (almost) equilibrated. This leads to the conclusion that the current dogma of density-dependent lymphocyte increase and return to normal (homeostatic) rates of renewal following depletion needs to be questioned.

Overall, this is an interesting study that addresses an important question of lymphocyte regeneration and turnover. The study robustly demonstrates that considerable dysregulation of lymphocyte production and loss rates can persist in the face of apparently normal (or near-normal) cell numbers. They present an interesting data set and use appropriate methods to analyze these data. The study is well written and the work is clearly presented. However, a more detailed and careful discussion with regard to the main conclusion, i.e., challenging the concept of homeostatic regulation, is suggested as this might not be directly inferred from the analysis.

In particular, the reviewers identified the following main aspects that should be addressed:

a) A more detailed discussion on the explanation and interpretation of the increased cell loss rates and the actual patient data is needed (especially addressing IL-7 levels)

b) The analysis of kinetically heterogeneous cell populations by accounting for Ki67 measurements in comparison to previous studies (Gossel et al., 2017, Hogan et al. *eLife* 2019) could be considered/discussed.

c) The general interpretation of challenging the concept of homeostatic regulation should be (re-)discussed in the light of possible caveats of the analysis.

These aspects will be subject to re-review to determine if these issues have been satisfactorily addressed. We would also like to point you to the specific points raised below that contain more detailed comments and summarize the individual reviews.

Essential revisions:

1) Interpretation:

a) The logical steps that lead from the observed increase in lymphocyte production/loss rates to the conclusion that these changes do not reflect a homeostatic response seem to be not that clear. It could be considered, that the data presented is in fact consistent with a "homeostatic response," in that lymphocyte production/loss rates are being modulated in an effort to normalise cell numbers. That these rates are both elevated compared to healthy controls likely indicates ongoing dysregulation of the homeostatic response in their patient cohort. One scenario that would be consistent with the data is an increase in cell loss rates (for example, damage to secondary lymphoid organs may affect delivery of cell survival/differentiation signals, as is discussed in the Discussion for B cells), resulting in a consequent increase of cell production rates as a homeostatic response to maintain cell numbers. The ongoing dysregulation of cell production/loss rates in these patients is in itself an important observation, given that cell numbers in many of the populations analysed have returned to normal or near-normal levels by the end of the study, which one might take at face value as an indication that homeostasis has been restored.

b) The assertion in the Discussion that there is a lack of evidence for homeostatic regulation of lymphocyte production and loss in humans is not justified. To give just one example to the contrary, there is a wealth of evidence that the classical homeostatic cytokine IL-7 regulates T cell numbers in human health and disease (see reviews by Barata, Durum and Seddon, 2019 and Lundstrom, Fewkes and Mackall, 2012).

2) IL-7 Measurements:

Despite citing an expected increase in plasma IL-7 levels in their patients, almost all measurements of IL-7 in patients are at the extreme low end of the normal range (Figure 2 and Figure 2—source data 1). The IQR seems to suspiciously start at 1, which is where the vast majority of the measurements are. Was the data not log base 10 transformed, but first one added and the log base 10 transformed? If so, the as the vast majority of patient samples seem to have no measurable IL-7 and, indeed, the suggestion seems to be that >25% of HC also have no measurable IL-7 either. What is the limit of detection in this assay? Could these patients possibly be deficient in IL-7? If so, this would have considerable implications on cell survival and proliferation rates. For the rest of the measurements in Figure 2, is the last sentence of the subsection “Heterogeneous T-cell reconstitution kinetics post-autoHSCT” not hard to justify? It's a challenge to statistically assess significance, but it would seem that the variance the patient samples is higher than that presented for the HCs, with several lying well outside IQRs. Perhaps they're within in 95% CIs?

3) Labelling:

a) The deuterium labelling regime used for the patients in this study was 3 weeks less than for the healthy controls. The reasons behind this choice are clearly stated in the Materials and methods section, and the impact of different median ages in each cohort is also addressed. However the possible implications of the reduced labelling period on the expected modelling results and how this might affect data interpretation are not discussed.

b) The authors make the important point that increased TREC frequencies in naive T cells should not be taken as evidence for increased thymic output, as recent thymic emigrants may be overrepresented in the reduced peripheral naive pool. Could this phenomenon also affect interpretation of their deuterium labelling studies? Developing thymocytes undergo many rounds of cell division, therefore could the DNA of recent thymic emigrants be enriched for deuterium (i.e., cells that underwent thymic development during the labelling phase and were subsequently exported to the periphery)?

c) The deuterium analysis establishes that cells are continuing to divide at a higher than base-line rate even when cell numbers have stabilized. Assuming differentiation is not the explanation, it's evident that increased levels of cell death, as inferred here through the modelling, is the only viable alternative. While no experimental mechanism that I know of exists to measure death rates, would it have been possible to, e.g., use DAPI staining to experimentally qualitatively substantiate the finding of increased death over controls? Perhaps there was insufficient material to do so, or this may be considered in a future study.

4) Cell heterogeneity/mathematical analysis/interpretation:

a) The multi-exponential model of kinetically distinct subsets is almost phenomenological as it may also represent temporal heterogeneity and all parameters are not identifiable. But if you assume there are two kinetically distinct subpopulations, then with the inclusion of Ki67 measurements was it not possible to constrain these parameters (subset sizes and division rates) more? Perhaps you still have the uncertainty of the partitioning between influx and cell division? One has to make assumptions about the Ki67 expression in immigrant cells.

b) Related to this – you quote the average production and loss rates. When fitting the bi-exponential model I think you find that the division rate and size of each memory subpopulation are highly correlated – but are there not invariant measures? Such as the proportion of per-cell production that derives from each (presumed) subset? (i.e. the product α_i_ p_i_) ?

c) It would be good to comment on the source of heterogeneity in memory. Might these two subsets broadly correspond to cells generated post-transplant (influx plus division) and residual old cells (division)? We have argued that newly generated memory T cells in mice have higher production and loss rates relative to more established populations (Hogan *eLife* 2019), and so maybe something similar is happening here – in the patients you are essentially seeing a mixture of memory dynamics seen in children (new memory) and adults (residual memory).

d) Similarly – naive T cells may also exhibit loss rates that decrease with cell age in mice (Rane et al., 2018 and Reynaldi et al., 2019) and in humans (Reu Plos Biol 2019) and the average Ki67 expression among naive T cells declines during the first 3 months of life. How do the elevated rates of production of naive T cells in these patients compare to those in children? Are you just seeing the natural behaviour of a naive T cell pool with a left-skewed age-distribution?

e) It has been observed that memory CD8 T cells do not appear to have a carrying capacity, at least in SPF mice – so perhaps there is a precedent here? In particular, maybe the CD4 memory deficiency is not too surprising? It has been shown that stable-ish memory CD4 numbers in adult SPF mice depend on early antigen exposure (Hogan *eLife* 2019), arguing against a carrying capacity/set point.

f) Are density effects only manifest when cell numbers are lower? Do you have Ki67 data from patients soon after transplant when numbers are increasing? Maybe in the Abstract you could emphasise that your results apply to quite a long time after transplant and so are perhaps especially surprising. Also – in the Discussion, you say memory production rates are not increased a few months after transplant. But they are probably increased soon after transplant?

5) Statistical methods:

a) As is noted in the Materials and methods, total leukocyte measurements are typically noisy. However, for pretty much every patient measurement, the point estimate (i.e. the measured number) is provided with what looks like no estimate of uncertainty or confidence. Just taking some individual samples, it would appear to be possibly significant. For example, consider Figure 6, patient F. In all panels, the seesaw numbers are consistent across the different compartments (day 500 low, day 520 high, day 550 low, etc.) suggestive of a systemic bias. Is no estimate of that uncertainty possible through, e.g., a control measurement of repeated measurements of samples from the same HC sample.

Comment: Not for this study as I presume it was not done, but perhaps something to think of for future studies: if you had measured HC samples as time-courses each time you measured a patient sample, you could manage the measurement uncertainty by treating each time-course (patient or healthy) as a single experimental observation and used one-sided permutation tests to establish statistical significance between the two groups.

b) In general, I would think it preferable to have a box plot rather than an IQR on each of the figures (IQR is not mentioned in Figure 3 legend, but I presume that's the case). It would take up no more space than you have, but give a better representation of the HC data, and provide a clearer communication of the inherent variability in the HC data.

---

## [Author Response]

Essential revisions:1) Interpretation:a) The logical steps that lead from the observed increase in lymphocyte production/loss rates to the conclusion that these changes do not reflect a homeostatic response seem to be not that clear. It could be considered, that the data presented is in fact consistent with a "homeostatic response," in that lymphocyte production/loss rates are being modulated in an effort to normalise cell numbers. That these rates are both elevated compared to healthy controls likely indicates ongoing dysregulation of the homeostatic response in their patient cohort. One scenario that would be consistent with the data is an increase in cell loss rates (for example, damage to secondary lymphoid organs may affect delivery of cell survival/differentiation signals, as is discussed in the Discussion for B cells), resulting in a consequent increase of cell production rates as a homeostatic response to maintain cell numbers. The ongoing dysregulation of cell production/loss rates in these patients is in itself an important observation, given that cell numbers in many of the populations analysed have returned to normal or near-normal levels by the end of the study, which one might take at face value as an indication that homeostasis has been restored.

We think our data at least counter the dogma that production rates go up when cell numbers are low, and normalize when cell numbers do. There are different possibilities that could explain the high levels of cell production and loss observed in our study. Loss rates may be increased due to increased cell death (e.g. as a result of damage to secondary lymphoid organs), or it may reflect increased homing to the tissues rather than cell death. We agree with the reviewers that we cannot rule out the possibility that increased production may be a homeostatic response to increased cell loss. Calling it “ongoing dysregulation” is indeed a good way of summarizing our findings, and therefore we have adopted the wording in the manuscript. We have also given more room for the different explanations in the text to get a more balanced discussion.

b) The assertion in the Discussion that there is a lack of evidence for homeostatic regulation of lymphocyte production and loss in humans is not justified. To give just one example to the contrary, there is a wealth of evidence that the classical homeostatic cytokine IL-7 regulates T cell numbers in human health and disease (see reviews by Barata, Durum and Seddon, 2019 and Lundstrom, Fewkes and Mackall, 2012).

In mice there is ample evidence for an in vivo role of IL-7, cytokines and other (membrane) proteins in (homeostatic control of) T cell proliferation and survival. For humans, the literature is more limited. We agree that IL-7 is one on the best-known cytokines implicated in homeostatic control of T cell numbers in humans. Based on this literature we also chose to include the measurements of IL-7 plasma levels in our study. In the revised version of the manuscript, we have added reference to this literature on IL-7 (see Discussion paragraph). However, studies testing the direct relationship between cell numbers and lymphocyte turnover in humans are much more limited and prone to interfering factors, such as infections and GvHD. The setting of autologous HSCT in a group of patients without clinical complications gave us the unique setting to investigate the direct link between cell numbers and lymphocyte production and loss rates. Based on the reviewers’ comments we now emphasize the finding that lymphocyte dynamics are still dysregulated when most subsets have normalized, and discuss possible explanations for this dysregulation more extensively.

2) IL-7 Measurements:Despite citing an expected increase in plasma IL-7 levels in their patients, almost all measurements of IL-7 in patients are at the extreme low end of the normal range (Figure 2 and Figure 2—source data 1). The IQR seems to suspiciously start at 1, which is where the vast majority of the measurements are. Was the data not log base 10 transformed, but first one added and the log base 10 transformed? If so, the as the vast majority of patient samples seem to have no measurable IL-7 and, indeed, the suggestion seems to be that >25% of HC also have no measurable IL-7 either. What is the limit of detection in this assay? Could these patients possibly be deficient in IL-7? If so, this would have considerable implications on cell survival and proliferation rates. For the rest of the measurements in Figure 2, is the last sentence of the subsection “Heterogeneous T-cell reconstitution kinetics post-autoHSCT” not hard to justify? It's a challenge to statistically assess significance, but it would seem that the variance the patient samples is higher than that presented for the HCs, with several lying well outside IQRs. Perhaps they're within in 95% CIs?

The limit of detection for IL-7 of the luminex assay performed in our study is 1.3pg/ml. We acknowledge that the variation in our assay is big and that the number of measurements is relatively small. To better represent the range of the healthy controls, we have included box plots showing the min to max range for the different plasma measurements in the revised version of the manuscript. As before, all samples with IL-7 levels below the detection level were set at 1 (this is now also mentioned in the legend of Figure 2).

Although we agree that there seems to be a tendency that IL-7 levels in patients were lower than in HC, there is no statistical support for this conclusion. This may be due to the relatively small sample size and the relatively large number of samples that fell below the limit of detection. Since the patients presented with CD4^+^ T-cell lymphopenia and significantly increased lymphocyte production rates, if anything we would have expected that they would have had increased levels of IL-7. Our data did not provide any evidence in this direction, however. Indeed, at the time of analysis (i.e. 200 days post-autoHSCT), patients did not have significantly higher levels of plasma IL-7 than healthy controls. We agree that these observations deserve more discussion in the manuscript, and have therefore included a paragraph in the Discussion to address this issue.

**Author response table 1. resptable1:** 

Patients	IL-7 (pg/ml)	Controls	IL-7 (pg/ml)
C-1	62.56	CTRL1	20.37
C-2	47.27	CTRL2	OOR <
C-3	17.08	CTRL3	81.35
F-1	OOR <	CTRL4	7.91
F-2	OOR <	CTRL5	113.81
F-3	OOR <	CTRL7	OOR <
A-1	OOR <	CTRL8	1.37
A-2	OOR <	CTRL9	13.21
A-3	OOR <	CTRL10	113.22
A-4	OOR <		
D-1	OOR <		
D-2	14.71		
D-4	39.94		
E-1	OOR <		
E-2	OOR <		
E-3	OOR <		
B-1	OOR <		
B-2	OOR <		

3) Labelling:a) The deuterium labelling regime used for the patients in this study was 3 weeks less than for the healthy controls. The reasons behind this choice are clearly stated in the Materials and methods section, and the impact of different median ages in each cohort is also addressed. However the possible implications of the reduced labelling period on the expected modelling results and how this might affect data interpretation are not discussed.

This has been addressed in the Materials and methods section of the revised version of the manuscript.

b) The authors make the important point that increased TREC frequencies in naive T cells should not be taken as evidence for increased thymic output, as recent thymic emigrants may be overrepresented in the reduced peripheral naive pool. Could this phenomenon also affect interpretation of their deuterium labelling studies? Developing thymocytes undergo many rounds of cell division, therefore could the DNA of recent thymic emigrants be enriched for deuterium (i.e., cells that underwent thymic development during the labelling phase and were subsequently exported to the periphery)?

The interpretation of the deuterium labelling results is not affected by this as the model already assumes that a fraction cS(t) of the DNA of cells that come from the thymus is labelled, i.e. we already assume that the DNA of RTE is enriched for deuterium.

c) The deuterium analysis establishes that cells are continuing to divide at a higher than base-line rate even when cell numbers have stabalized. Assuming differentiation is not the explanation, it's evident that increased levels of cell death, as inferred here through the modelling, is the only viable alternative. While no experimental mechanism that I know of exists to measure death rates, would it have been possible to, e.g., use DAPI staining to experimentally qualitatively substantiate the finding of increased death over controls? Perhaps there was insufficient material to do so, or this may be considered in a future study.

We agree that it would have been extremely valuable to have incorporated cell death markers, such as DAPI or Annexin V, in our assay. However, at the time of study design we did not take into account that cell death would play such an important role in our study. Several published studies have analysed T-cell survival following allogeneic HSCT and found that the fraction of pro-apoptotic cells increases following transplantation. We have added this information to the Discussion of the revised manuscript. In future studies we will certainly take death markers along.

4) Cell heterogeneity/mathematical analysis/interpretation:a) The multi-exponential model of kinetically distinct subsets is almost phenomenological as it may also represent temporal heterogeneity and all parameters are not identifiable. But if you assume there are two kinetically distinct subpopulations, then with the inclusion of Ki67 measurements was it not possible to constrain these parameters (subset sizes and division rates) more? Perhaps you still have the uncertainty of the partitioning between influx and cell division? One has to make assumptions about the Ki67 expression in immigrant cells.

The multi-exponential model was used because it yields lymphocyte turnover rates that are independent of the length of the labelling period, as demonstrated before by Westera et al., 2013. A draw-back of the multi-exponential model is indeed that not all parameters may be identifiable. We therefore only interpret the average turnover rates, which tend to have much smaller confidence intervals than the individual parameters. Unfortunately, we have no clue which sub-populations constitute the kinetically heterogeneous lymphocyte populations, and we do not see how Ki67 data could help to constrain these parameters. Since Ki67 is a snapshot marker, even cells from a subpopulation with rapid turnover may be Ki67-negative at the moment of our measurement. Even Ki67 measurements will therefore only provide information about the average proliferation rate in a cell population, and not about the proliferation rates of the underlying subsets. Although it is a very interesting thought, we think that using the Ki67 data to constrain parameter estimates based on deuterium labelling would only work if the subpopulations that together constitute a heterogeneous lymphocyte population i) have extremely different kinetics, e.g. one subpopulation without turnover and one subpopulation with considerable turnover, and ii) happen to be identifiable based on the relatively low number of markers that were taken along in our FACS analyses. As mentioned by the reviewer, the fact that deuterium labelling can be due to both influx and cell division complicates this even further.

b) Related to this – you quote the average production and loss rates. When fitting the bi-exponential model I think you find that the division rate and size of each memory subpopulation are highly correlated – but are there not invariant measures? Such as the proportion of per-cell production that derives from each (presumed) subset? (i.e. the product α_i_ p_i_) ?

We very much like this idea, and have tested it on the different lymphocyte populations that we have analysed. Surprisingly, we typically find that even the confidence intervals on the product α_i_p_i_ can be large, and considerably larger than the confidence interval on the average production rate. See for an example Author response image 1 in which we plotted the spread (represented by the quartiles) on the different parameters of the memory CD4 T-cell population of one individual. The values of the best fit were scaled to 1, so that the spread on the different parameters and parameter combinations can be compared. Since the confidence intervals on α_i_p_i_ (in contrast to those on the average production rates) were not consistently smaller than those on the individual parameters α_i_ and p_i._ we have decided not to include this in our revised manuscript, for the same reasons as why we do not report the individual parameters α_i_ and p_i._

c) It would be good to comment on the source of heterogeneity in memory. Might these two subsets broadly correspond to cells generated post-transplant (influx plus division) and residual old cells (division)? We have argued that newly generated memory T cells in mice have higher production and loss rates relative to more established populations (Hogan eLife 2019), and so maybe something similar is happening here – in the patients you are essentially seeing a mixture of memory dynamics seen in children (new memory) and adults (residual memory).

We agree that an altered contribution of young and old cells with different production and loss rates would be one of the possible explanations of our findings. We have included this in the Discussion of the revised manuscript.

d) Similarly – naive T cells may also exhibit loss rates that decrease with cell age in mice (Rane et al., 2018 and Reynaldi et al., 2019) and in humans (Reu Plos Biol 2019) and the average Ki67 expression among naive T cells declines during the first 3 months of life. How do the elevated rates of production of naive T cells in these patients compare to those in children? Are you just seeing the natural behaviour of a naive T cell pool with a left-skewed age-distribution?

We agree that also for naive T cells the increased production and loss rates that we observed in HSCT patients may reflect the relatively large contribution of relatively young cells. Increased lymphocyte production rates after HSCT may thus be a reflection of a “younger” immune system. We have added this interpretation, as well as a reference to the literature on this to the Discussion of the revised manuscript.

Unfortunately, there are no estimates of lymphocyte turnover in children based on deuterium labelling to compare to. Strictly speaking, we therefore do not know whether the observed differences in deuterium labelling between HSCT patients and healthy controls are fully explained by a left-skewed age-distribution of lymphocytes in HSCT patients. When comparing Ki-67 expression levels of naive and memory T cells between HSCT patients and young healthy children (van Gent et al., 2009), we found very similar levels. This suggests that in HSCT patients we may thus indeed see the natural dynamics of a T-cell pool with a left-skewed age-distribution. Our TREC data would also be in line with this interpretation. We have therefore added this as a possible explanation for our findings in the Discussion of the revised manuscript. Unfortunately, the study of Reu et al. only has data from the age of 20 years onwards. We have therefore not included this paper in this part of the Discussion.

e) It has been observed that memory CD8 T cells do not appear to have a carrying capacity, at least in SPF mice – so perhaps there is a precedent here? In particular, maybe the CD4 memory deficiency is not too surprising? It has been shown that stable-ish memory CD4 numbers in adult SPF mice depend on early antigen exposure (Hogan eLife 2019), arguing against a carrying capacity/set point.

Although we very much like this point raised by the reviewer, we feel that we would complicate the paper enormously if we start discussing this in the absence of any data in humans to support this interpretation. It would be great if in the future we could find a condition in which to test this hypothesis in humans.

f) Are density effects only manifest when cell numbers are lower? Do you have Ki67 data from patients soon after transplant when numbers are increasing? Maybe in the Abstract you could emphasise that your results apply to quite a long time after transplant and so are perhaps especially surprising. Also – in the Discussion, you say memory production rates are not increased a few months after transplant. But they are probably increased soon after transplant?

Unfortunately, we had no blood samples from the participants of our study earlier after transplant. Malphettes, 2003 studied adults with multiple myeloma who received an autologous transplantation and reported the following:

“At month 1, 19% of the CD4^+^ and 24% of the CD8^+^ T cells (CD45RO+) exhibit the Ki67 antigen. These percentages decreased as soon as the third month after grafting and remained at values slightly above normal range (2%-5%) until month 15 without any difference between the CD4^+^ and CD8^+^”.

The Ki-67 values observed in the study by Malphettes et al. >3months after transplantation seem to be in line with the observations from our study. Therefore, we think that it is likely that shortly after transplant (<3 months) Ki-67 expression by memory T cells may well have been increased. We have followed your suggestion in the Abstract and briefly refer to Malphettes’ findings in the Discussion of the revised manuscript.

5) Statistical methods:a) As is noted in the Materials and methods, total leukocyte measurements are typically noisy. However, for pretty much every patient measurement, the point estimate (i.e. the measured number) is provided with what looks like no estimate of uncertainty or confidence. Just taking some individual samples, it would appear to be possibly significant. For example, consider Figure 6, patient F. In all panels, the seesaw numbers are consistent across the different compartments (day 500 low, day 520 high, day 550 low, etc.) suggestive of a systemic bias. Is no estimate of that uncertainty possible through, e.g., a control measurement of repeated measurements of samples from the same HC sample.Comment: Not for this study as I presume it was not done, but perhaps something to think of for future studies: if you had measured HC samples as time-courses each time you measured a patient sample, you could manage the measurement uncertainty by treating each time-course (patient or healthy) as a single experimental observation and used one-sided permutation tests to establish statistical significance between the two groups.

Leukocyte numbers in blood are typically highly variable. Such variability may to a small extent come from measurement error, but is mainly thought to be due to real differences in leukocyte numbers in the blood at different time points. Many variables, e.g. exercise or moment of the day, may influence the measurement. Repeated measurements of samples from healthy controls would therefore not help to reduce this variation. It is exactly for this reason that we chose to describe total leukocyte numbers using linear regression, to then calculate cell numbers in each subset using the percentages of cells in the different gates. We have clarified this in the revised manuscript (see Materials and methods). Moreover we have indicated in the figure legends when this correction was done for the data shown in the figure.

b) In general, I would think it preferable to have a box plot rather than an IQR on each of the figures (IQR is not mentioned in Figure 3 legend, but I presume that's the case). It would take up no more space than you have, but give a better representation of the HC data, and provide a clearer communication of the inherent variability in the HC data.

We agree that showing the data as a box plot may help clarify the min-max range of healthy control values. We have changed Figures 2, 3, 6, and Figure 5—figure supplement 1 and their corresponding figure legends accordingly.